# Learning to Compress Prompts with Gist Tokens

**Jesse Mu, Xiang Lisa Li, Noah Goodman**
Stanford University
muj@cs.stanford.edu, {xlisali,ngoodman}@stanford.edu

## Abstract

Prompting is the primary way to utilize the multitask capabilities of language models (LMs), but prompts occupy valuable space in the input context window, and repeatedly encoding the same prompt is computationally inefficient. Finetuning and distillation methods allow for specialization of LMs without prompting, but require retraining the model for each task. To avoid this trade-off entirely, we present *gisting*, which trains an LM to compress prompts into smaller sets of "gist" tokens which can be cached and reused for compute efficiency. Gist models can be trained with no additional cost over standard instruction finetuning by simply modifying Transformer attention masks to encourage prompt compression. On decoder (LLaMA-7B) and encoder-decoder (FLAN-T5-XXL) LMs, gisting enables up to 26x compression of prompts, resulting in up to 40% FLOPs reductions, 4.2% wall time speedups, and storage savings, all with minimal loss in output quality.

## 1 Introduction

Consider the prompt of a Transformer [34] language model (LM) like ChatGPT:[1]

```
You are ChatGPT, a large language model trained by OpenAI. You answer as concisely as
possible for each response (e.g. don't be verbose). It is very important that you answer
as concisely as possible, so please remember this. If you are generating a list, do not
have too many items. Keep the number of items short.
Knowledge cutoff: 2021-09 Current date: <TODAY>
```

With millions of queries a day, an unoptimized ChatGPT would encode this prompt over and over with a self-attention mechanism whose time and memory complexity is quadratic in the length of the input. Caching the Transformer activations of the prompt can prevent some recomputation, yet this strategy still incurs memory and storage costs as the number of cached prompts grows. At large scales, even small reductions in prompt length could lead to substantial compute, memory, and storage savings over time, while also letting users fit more content into an LM's limited context window.

How might we reduce the cost of this prompt? One typical approach is to finetune or distill [1, 30] the model to behave similarly to the original model without the prompt, perhaps with parameter-efficient adaptation methods [15, 16, 19]. Yet a fundamental drawback of this approach is that it requires retraining the model for each new prompt (Figure 1, bottom left).

Instead, we propose **gisting** (Figure 1, top right), which compresses arbitrary prompts into a smaller set of Transformer activations on top of virtual "gist" tokens, *a la* prefix-tuning [19]. But where prefix-tuning requires learning prefixes via gradient descent for each task, gisting adopts a meta-learning approach, where we simply predict the gist prefixes zero-shot given only the prompt, allowing for generalization to unseen instructions without any additional training. Since gist tokens are much shorter than the full prompt, gisting allows arbitrary prompts to be compressed, cached, and reused for compute efficiency.

---

[1]reddit.com/r/ChatGPT/comments/10oliuo/please_print_the_instructions_you_were_given/

37th Conference on Neural Information Processing Systems (NeurIPS 2023).

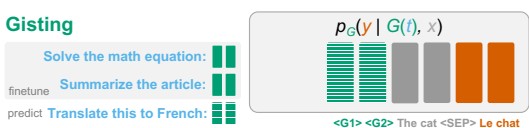

Figure 1: **Prompting** retains the multitask capabilities of LMs, but is inefficient. **Finetuning/distillation** is more efficient, but requires training a model for each task. **Gisting** compresses prompts into activations on top of "gist tokens", saving compute and generalizing to novel tasks at test time. Each vertical rectangle represents a stack of Transformer activations.

In this paper, we further propose a very simple way to learn a gist model: doing instruction tuning [38] with gist tokens inserted after the prompt, and a modified attention mask preventing tokens *after* the gist tokens from attending to tokens *before* the gist tokens. This allows a model to learn prompt compression and instruction following at the same time, with no additional training cost.

On decoder-only (LLaMA-7B) and encoder-decoder (FLAN-T5-XXL) LMs, gisting achieves prompt compression rates of up to **26x**, while maintaining output quality similar to the original models in human evaluations. This results in up to 40% FLOPs reduction and 4.2% latency speedups during inference, with greatly decreased storage costs compared to traditional prompt caching approaches.

## 2   Gisting

We will first describe gisting in the context of instruction finetuning [38]. We have an instruction-following dataset $\mathcal{D} = \{(t_i, x_i, y_i)\}_{i=1}^{N}$, where $t$ is a task encoded with a natural language prompt (e.g. Translate this to French), $x$ is an (optional) input for the task (e.g. The cat), and $y$ is the desired output (e.g. Le chat). Given a (usually pretrained) LM, the aim of instruction finetuning is to learn a distribution $p_{\text{LM}}(y \mid t, x)$, typically by concatenating $t$ and $x$, then having the LM autoregressively predict $y$. At inference time, we can *prompt* the model with a novel task $t$ and input $x$, decoding from the model to obtain its prediction.

However, this pattern of concatenating $t$ and $x$ has drawbacks: Transformer LMs have limited context windows, bounded either by architecture or memory limits. Furthermore, given that attention scales quadratically in the length of the input, long prompts $t$, especially those that are repeatedly reused, are computationally inefficient. What options do we have to reduce the cost of prompting?

One simple option is to finetune the LM for a *specific* task $t$. That is, given $\mathcal{D}^t = \{(x_i, y_i)\}_{i=1}^{N^t}$, the dataset containing input/output examples only under task $t$, we can learn a specialized LM $p_{\text{LM}}^t(y \mid x)$ which is faster because it does not condition on $t$. Parameter-efficient finetuning methods such as prefix-/prompt-tuning [18, 19] or adapters [15, 16] promise to do so at a fraction of the cost of full finetuning, and newer methods like HyperTuning [25] eliminate gradient descent entirely, instead predicting the parameters of the specialized model directly from $\mathcal{D}^t$. Yet problems with these methods still remain: we must store at least a subset of model weights for each task, and more importantly, for each task $t$, we must collect a corresponding dataset of input/output pairs $\mathcal{D}^t$ to adapt the model.

Gisting is a different approach that amortizes both (1) the inference-time cost of prompting $p_{\text{LM}}$ with $t$ and (2) the train-time cost of learning a new $p_{\text{LM}}^t$ for each $t$. The idea is to learn a *compressed* version of $t$, $G(t)$, such that inference from $p_G(y \mid G(t), x)$ is faster than $p_{\text{LM}}(y \mid t, x)$. In LM terms, $G(t)$ will be the key/value activations on top a set of *gist tokens*, smaller than the number of tokens in $t$, yet still inducing similar behavior from the LM. Also known as a Transformer *prefix* [19], $G(t)$ can then be cached and reused for compute efficiency. Crucially, we expect $G$ to generalize to unseen tasks: given a new task $t$, we can predict and use the gist activations $G(t)$ *without any additional training*.

### 2.1   A Context Distillation Perspective

An alternative way to view gisting is through the lens of distillation of an already instruction-tuned LM $p_{\text{LM}}(y \mid t, x)$. Askell et al. [1] and Snell et al. [30] define *context distillation* as the process of finetuning a new LM $p_{\text{CD}}^t$ to mimic the original LM without the prompt ("context") $t$, via the loss

$$\mathcal{L}_{\text{CD}}(p_{\text{CD}}^t, t) = \mathbb{E}_x \left[ D_{\text{KL}}(p_{\text{LM}}(y \mid t, x) \parallel p_{\text{CD}}^t(y \mid x)) \right]. \tag{1}$$

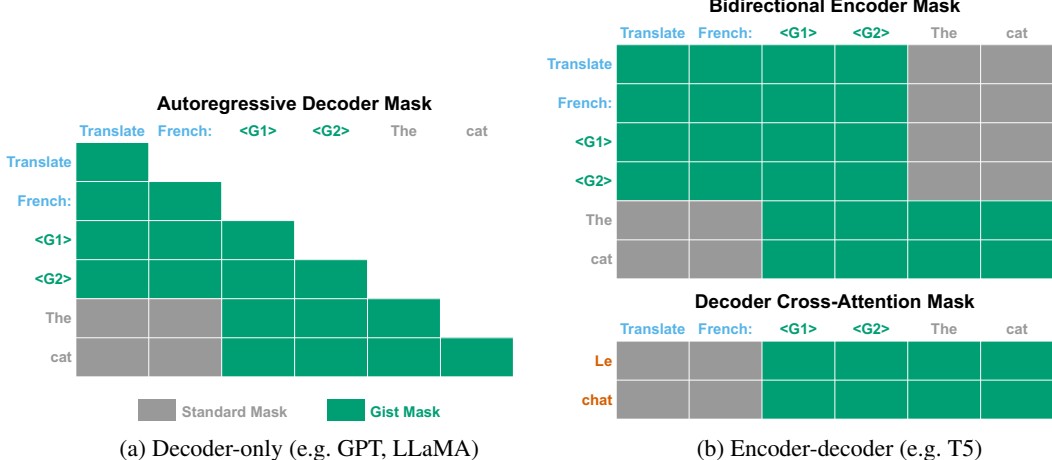

Figure 2: **Gist Masking**. Attention mask modifications for (a) decoder-only and (b) encoder-decoder Transformer LMs to encourage prompt compression into gist tokens `<G1>` `<G2>`. In these tables, cell $(r, c)$ shows whether token $r$ can attend to token $c$ during self- or cross-attention.

The insight to be gained from this perspective is that we do not need any external data $\mathcal{D}$: this KL objective can be approximated by finetuning $p_{\text{CD}}^t$ on a synthetic sampled dataset $\hat{\mathcal{D}}^t = \{(\hat{x}_i, \hat{y}_i)\}$ where $(\hat{x}_i, \hat{y}_i) \sim p_{\text{LM}}(\cdot \mid t)$. This is precisely the approach taken by recent work [1, 7, 30], including Wingate et al. [40], who notably learn to compress a single discrete prompt into a soft prompt via gradient descent, similar to this paper.

However, we differ from this prior work in that we are not interested in distilling just a single task, but in amortizing the cost of distillation across a *distribution* of tasks $T$. That is, given a task $t \sim T$, instead of obtaining the distilled model via gradient descent, we use $G$ to simply *predict* the gist tokens ($\approx$ parameters) of the distilled model, in the style of HyperNetworks [13] and HyperTuning [25]. Our "meta" distillation objective is thus (with changes highlighted in **blue**):

$$\mathcal{L}_G(p_G, T) = \mathbb{E}_{t \sim T, x} \left[ D_{\text{KL}}(p_{\text{LM}}(y \mid t, x) \parallel p_G(y \mid G(t), x)) \right]. \tag{2}$$

In the experiments we describe below, we train on synthetic instruction-following data sampled from instruction-tuned variants of GPT-3 [3, 23]. Thus, these experiments can indeed be seen as a form of context distillation for the GPT-3 series models.

## 3 Learning Gisting by Masking

Having just described the general framework of gisting, here we will explore an extremely simple way of learning such a model: using the LM itself as the gist predictor $G$. This not only leverages the pre-existing knowledge in the LM, but also allows us to learn gisting by simply doing standard instruction finetuning while modifying the Transformer attention masks to enforce prompt compression. This means that gisting incurs *no* additional training cost on top of standard instruction finetuning!

Specifically, we add a *single* gist token $g$ to the model vocabulary and embedding matrix. Then, given a (task, input) pair $(t, x)$, we concatenate $t$ and $x$ with a set of $k$ copies of $g$ in between: $(t, g_1, \ldots, g_k, x)$, e.g. `Translate French: <G1> <G2> The cat`.[2] The model is then restricted such that input tokens *after* the gist tokens cannot attend to any of the prompt tokens *before* the gist tokens (but they *can* attend to the gist tokens). This forces the model to compress the prompt information into the gist prefix, since the input $x$ (and output $y$) cannot attend to the prompt $t$.

Figure 2 illustrates the required changes. For **decoder-only** LMs such as GPT-3 [3] or LLaMA [33] that normally admit an autoregressive, causal attention mask, we simply mask out the lower-left corner of the triangle (Figure 2a). For **encoder-decoder** LMs (e.g. T5; [28]) with a bidirectional encoder followed by an autoregressive decoder, two changes are needed (Figure 2b). First, in the encoder, which normally has no masking, we prevent the input $x$ from attending to the prompt $t$. But

---

[2]Again, the gist token is the same from $g_1$ to $g_k$; what changes is the activations on top of each token.

we must also prevent the prompt $t$ and gist tokens $g_i$ from attending to the input $x$, since otherwise the encoder learns different representations depending on the input. Finally, the decoder operates as normal, except during cross-attention, we prevent the decoder from attending to the prompt $t$.

Overall, these masking changes are extremely simple and can be implemented in roughly 10 source lines of code. See Appendix A for a sample PyTorch implementation which can be used as a drop-in replacement for attention masking in deep learning libraries such as Hugging Face Transformers [41].

# 4 Experiments

## 4.1 Data

A dataset with a large variety of tasks (prompts) is crucial to learn gist models that can generalize. To obtain the largest possible set of tasks for instruction finetuning, we create a dataset called Alpaca+, which combines the Self-Instruct [36] and Stanford Alpaca [31] instruction tuning datasets, each consisting of $(t, x, y)$ tuples sampled from OpenAI's `text-davinci-001` and `text-davinci-003` variants of GPT-3, respectively. In total, Alpaca+ has 130,321 examples, with 104,664 unique tasks $t$, 48,530 unique inputs $x$, and anywhere from 0–5 inputs per task (0.64 on average).

Note that ~59% of tasks in Alpaca+ have no inputs (e.g. `Write me a poem about frogs`), in which case we simply omit the input $x$. While it is less interesting to cache such prompts since they are not input-dependent, they still serve as valuable training signal for learning prompt compression. Overall, while Alpaca+ is noisy and imperfect, Wang et al. [36] and Taori et al. [31] nevertheless show that models trained on such data achieve comparable performance to the original models from which the data is sampled, making this a promising testbed for studying gisting.

From Alpaca+ we hold out 3 validation splits: 1000 **Seen** prompts (with unseen, non-empty inputs); 1000 **Unseen** prompts (with non-empty inputs); and the 252 hand-written **Human** prompts and completions used in Wang et al. [36], of which 83% have non-empty inputs. The latter two splits test generalization to unseen instructions, with the **Human** split posing a stronger out-of-distribution (OOD) challenge: the average training prompt has ~20 tokens, compared to ~26 in the human split.

## 4.2 Models

To demonstrate gisting across multiple Transformer LM architectures, we experiment with LLaMA-7B [33], a decoder-only GPT-style model with ~7B parameters, and FLAN-T5-XXL [8], an encoder-decoder T5 model [28] with 11B parameters. For each of these models, we train models with a varying number of gist tokens $k \in \{1, 2, 5, 10\}$, using the modified attention masks described in Section 3. To assess how the model is learning prompt compression, we calibrate performance against upper- and lower-bound baselines and a simple discrete compression strategy:

**Positive Control.** As an upper bound on performance, we train a model with a single gist token, but without any modifications the attention mask. This is akin to doing standard instruction finetuning.

**Negative Control.** As a lower bound on performance, we train a model without access to the task $t$. This is similar to a "random gist token" baseline, which allows us to measure how the model would do if it failed to compress *any* information into the gist prefix.

**Discrete Compression with TF-IDF.** An alternative approach to compression is simply using fewer discrete tokens to express the same task. Achieving compression rates similar to gisting, however, requires compression far beyond any threshold of fluency. Nevertheless, as a baseline, we compute TF-IDF statistics over the set of instructions in the Alpaca+ training set to extract the most relevant keyword in each instruction. Some examples from the training set include (see Appendix G for more):

```
Write a letter to your boss asking for an increase in salary → salary
Given two integers, find their average → average
```

We then replace each instruction in Alpaca+ with the first subword token from each keyword, resulting in compression rates equivalent to a model trained with a single gist token. Similarly to the positive control, we do standard instruction finetuning over Alpaca+ with these compressed instructions.

For full training, data, and compute details, and a link to code, see Appendix B.

### 4.3 Evaluation

Our evaluation uses a combination of automated metrics and AI- and human-assisted evaluation:

**ROUGE-L.** We first use ROUGE-L, a simple lexical overlap statistic [20], used in previous open-ended instruction finetuning work [37, 38]. The `text-davinci-{001,003}` completions are used as references, except for the Human split, where we use the gold-standard human reference.

**ChatGPT.** Next, we use ChatGPT-3.5 [22] to compare the outputs of our models to the positive control. While this is an imperfect metric, it allows for much faster and cheaper evaluation than human experiments, with an arguably more meaningful semantic signal than ROUGE-L. Recent work has found that ChatGPT can be used for text annotation and evaluation [12, 17, 35] with near-human performance, and similar model-based evaluations have been conducted with recent LMs [5, 11, 31].

Specifically, given a task $t$, input $x$, and outputs from two models $(y_1, y_2)$ identified only as Assistants A and B, ChatGPT was asked to choose which assistant response is better, explaining its reasoning in Chain-of-Thought fashion [39]. If the models produced the same output, or were equally bad, ChatGPT was allowed to call a tie. We gave examples of desired outputs in ChatGPT's prompt, and randomized the order of presentation between the models for each query to avoid order effects. The full prompt given to ChatGPT and evaluation details are in Appendix C. Using these outputs, we measure the win rate of a model against the positive control: a win rate of 50% indicates that the model is of comparable quality to a model that does no prompt compression.

**Human eval.** Finally, after prototyping with ChatGPT, we select the best gist compression models and do a Human evaluation on a random subset of 100 of the 252 examples in the Human validation split. For each of the 100 examples, we recruited 3 US or UK-based, English-fluent annotators from Prolific, and asked them to rate model outputs in the same style as the ChatGPT evaluation above (see Appendix D for full details, including the annotation interface). The only difference is that human participants were allowed to select "I Don't Know" in cases where they had inadequate domain knowledge to accurately judge the responses, e.g. if the question was a coding question; we drop these responses (~10%) during analysis. With this human evaluation, we are not only interested in evaluating our final models, but also validating whether ChatGPT can be used as a reliable replacement for human annotation on this task.

## 5 Results

ROUGE-L and ChatGPT evaluations for LLaMA-7B and FLAN-T5-XXL, with varying numbers of gist tokens, are shown in Figure 3. Models were generally insensitive to the number of gist tokens $k$: compressing prompts into a single token prefix did not substantially underperform larger prefixes. In fact, having too many gist tokens hurts performance in some cases (e.g. LLaMA-7B, 10 gist tokens), perhaps because the increased capacity enables overfitting to the training distribution. Thus, we use the single gist token models for the rest of the experiments in the paper, and report the exact numbers for the single-token models, with the positive, negative, and TF-IDF baselines, in Table 1.

On **Seen** instructions, gist models attain near-identical ROUGE and ChatGPT performance as their positive control models (48.6% and 50.8% win rates for LLaMA-7B and FLAN-T5-XXL, respectively). But we are most interested in generalization to unseen tasks, as measured by the other two splits. On **Unseen** prompts within the Alpaca+ distribution, we again see competitive performance: 49.7% (LLaMA) and 46.2% (FLAN-T5) win rates against the positive controls. It is on the most challenging OOD **Human** split where we see slight drops in win rate to 45.8% (LLaMA) and 42.5% (FLAN-T5), though these numbers are still quite competitive with the positive control. Finally, gist compression is vastly superior to discrete compression; the TF-IDF models in Table 1 only marginally outperform the negative control models across the board.

Table 2 shows the human evaluation results on the Human validation split, comparing the single gist token models to the positive control. Overall, human annotators agree with ChatGPT, with average win rates of 52.3% (vs. 48.0%) for LLaMA-7B and 40.6% (vs. 42.0%) for FLAN-T5-XXL. Importantly, this agreement persists at the level of individual responses. The average pairwise Cohen's $\kappa$ among human annotators is .24 for LLaMA-7B and .33 for FLAN-T5-XXL. Because humans will often arbitrarily choose one response over another even for samples of equal quality, these numbers

Table 1: **Results for single gist tokens.** ROUGE-L and ChatGPT scores for Gist, TF-IDF, and positive/negative controls. Parentheses are scores normalized between positive/negative controls.

| Model | | Seen | | Unseen | | Human | |
|---|---|---|---|---|---|---|---|
| | | ROUGE-L | ChatGPT % | ROUGE-L | ChatGPT % | ROUGE-L | ChatGPT % |
| LLaMA-7B | Pos | 58.0 (100) | 50.0 (100) | 48.1 (100) | 50.0 (100) | 27.0 (100) | 50.0 (100) |
| | Gist | 57.8 (99.2) | 48.6 (92.4) | 46.6 (91.0) | 49.7 (98.8) | 23.9 (75.4) | 45.8 (84.9) |
| | TF-IDF | 38.1 (24.5) | 34.5 (16.2) | 34.0 (15.6) | 29.3 (15.9) | 16.5 (16.7) | 24.6 (8.6) |
| | Neg | 31.5 (0) | 31.5 (0) | 31.4 (0) | 25.4 (0) | 14.4 (0) | 22.2 (0) |
| FLAN-T5-XXL | Pos | 50.6 (100) | 50.0 (100) | 45.7 (100) | 50.0 (100) | 23.9 (100) | 50.0 (100) |
| | Gist | 48.9 (93.2) | 50.8 (103.9) | 43.8 (88.6) | 46.2 (84.4) | 21.7 (80.9) | 42.5 (63.2) |
| | TF-IDF | 32.0 (25.9) | 35.9 (30.5) | 34.3 (31.3) | 31.0 (22.1) | 13.5 (9.6) | 28.4 (-5.9) |
| | Neg | 25.5 (0) | 29.7 (0) | 29.1 (0) | 25.6 (0) | 12.4 (0) | 29.6 (0) |

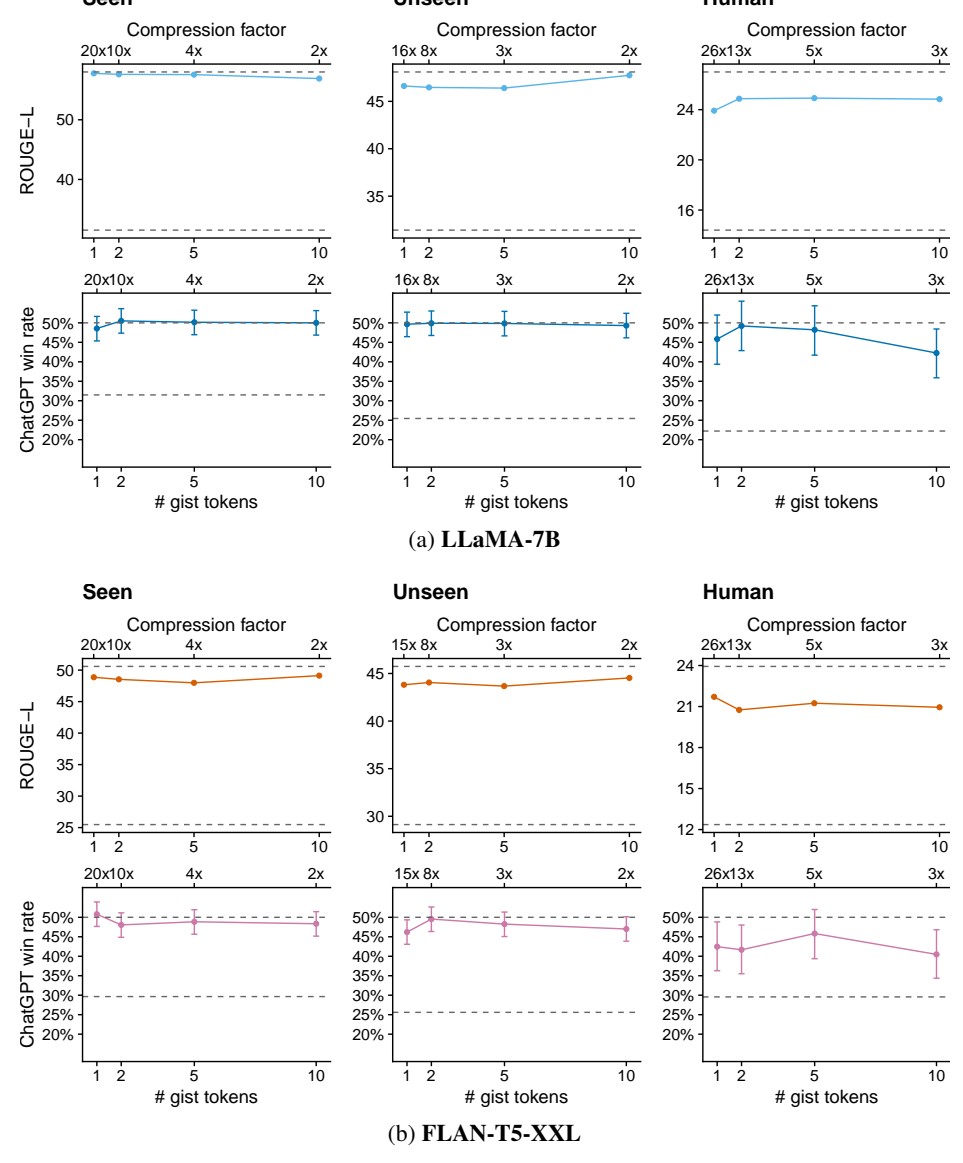

(a) **LLaMA-7B**

(b) **FLAN-T5-XXL**

Figure 3: **Varying the number of gist tokens**. ROUGE-L and ChatGPT scores for (a) **LLaMA-7B** and (b) **FLAN-T5-XXL** for different gist tokens. Dashed lines indicate positive and negative control performance. Error bars are 95% exact binomial confidence intervals, splitting ties equally between models [10] and rounding down in favor of the positive control in case of an odd number of ties. Compression factors are calculated by computing the average token length of the validation split and dividing by the number of gist tokens.

Table 2: **Human evaluation results.** Win rate and inter-annotator agreement of single token gist models over positive control according to 3 Human annotators (H1–H3), their average, and ChatGPT (95% confidence intervals in parentheses), for 100 out of 252 examples in the Human validation split.

| | Gist Win % over Pos | | | | | Agreement (Cohen's $\kappa$) | |
|---|---|---|---|---|---|---|---|
| Model | H1 | H2 | H3 | Human (H1–H3) | ChatGPT | Human | ChatGPT |
| LLaMA-7B | 51.1 | 44.5 | 59.8 | 52.3 (46.1, 58.4) | 48.0 (38.0, 58.2) | .24 | .29 |
| FLAN-T5-XXL | 43.0 | 41.9 | 37.2 | 40.6 (34.6, 46.8) | 42.0 (32.2, 52.3) | .33 | .29 |

are fairly low; however, ChatGPT shows similar levels of agreement, with average $\kappa$ across each of the 3 human annotators at .29 for both models. These results, paired with the similar overall win rates, show that using ChatGPT is similar to simply recruiting an additional human annotator, and corroborates the broader results in Figure 3. See Appendix D for more human evaluation results, including a breakdown of agreement across annotators, and Appendix G for examples of instructions, model outputs, and human/ChatGPT judgments in the Human validation split.

Since our aim is having the gist models mimic the original models, one might ask how often the gist model is identical to the positive control. Figure A.3 in Appendix E shows how often this happens: for **Seen** tasks (but unseen inputs), the gist model outputs exactly match the positive control nearly 50% of the time. This drops to ~20–25% for **Unseen** tasks and ~10% for the OOD **Human** tasks.

Overall, our results show that gist models can reliably compress prompts, even to some degree those that lie outside the training distribution, especially for decoder-only LMs (LLaMA). Encoder-decoder LMs like FLAN-T5 show slightly worse OOD performance; one possible reason is that gist masking inhibits the bidirectional attention flow in the encoder, which is more challenging to adjust to than simply limiting the history of a decoder LM. Future work will be needed to investigate this hypothesis.

## 5.1 Failure cases

While the responses from the gist models are often indistinguishable from the positive control, we nevertheless do see failure cases where the gist token models do not perfectly compress the prompt. We give two examples where this occurs in the Human validation split: first, when the instruction contains **specific details**, e.g. phrases that must be copied verbatim into the output (in the snippets below, the names of Assistants A/B are hidden to ChatGPT):

```
Instruction: Give the news title a category. Pick a category from the list of News &
Buzz, Travel, Style, Arts & Culture, Politics, Tech, and Science & Health. <G1>
Input: The #Banksy Exhibit in Cambridge, MA is absolutely terrific.
Assistant A (FLAN-T5-XXL gist): Culture
Assistant B (pos control): Arts & Culture

ChatGPT choice: B
ChatGPT reason: The news title is about an art exhibit, so Assistant B's response of
'Arts & Culture' is more accurate and specific than Assistant A's 'Culture'.
```

We also see more cases of **runaway generations** that do not occur with the positive control model:

```
Instruction: Provide an example of how a table of contents can be generated automatically
in a LATEX document. Make sure your example contains the necessary commands. <G1>
Assistant A (pos control): \tableofcontents \chapter{Introduction} \chapter{Theory}
\chapter {Experiments} \chapter{Conclusion}
Assistant B (LLaMA-7B gist): \tableofcontents \chapter{Introduction} \section{The Basics}
\subsection {The Basics} \subsubsection{The Basics} \paragraph{The Basics}
\subparagraph {The Basics} [repeats for hundreds of tokens] \subparagraph{The Basics}

ChatGPT choice: A
ChatGPT reason: Assistant A provides a simple and correct example of generating a table
of contents in LaTeX. Assistant B's example is unnecessarily long and does not follow
the standard structure of a table of contents.
```

While it is unclear why only the gist models exhibit this behavior, these issues can likely be mitigated with more careful sampling techniques.

Table 3: **Gist efficiency improvements**. For different caching strategies (**None**, **Instruction**, **Gist**), we record CUDA wall time and GFLOPs (± std dev). Then we report the absolute/relative improvement of **Gist Caching** over these alternative strategies, with 95% confidence intervals in parentheses.

| Model | Metric | Caching Strategy | | | Absolute/Relative Δ | |
|---|---|---|---|---|---|---|
| | | None | Instruction[i] | Gist[ii] | vs. None | vs. Instruction |
| LLaMA-7B | CUDA time (ms) ↓ | 23.4 ± 6.88 | 22.1 ± 6.58 | **21.8 ± 6.55** | 1.60 (1.29, 1.90) 6.8% (5.5, 8.1) | .221 (.140, .302) 1.0% (.63, 1.4) |
| | GFLOPs ↓ | 915 ± 936 | 553 ± 900 | **552 ± 899** | 362 (337, 387) 40% (37, 42) | .607 (.448, .766) .11% (.08, .14) |
| FLAN-T5-XXL | CUDA time (ms) ↓ | 31.0 ± 5.31 | N/A | **29.7 ± 5.07** | 1.30 (1.10, 1.51) 4.2% (3.6, 4.9) | N/A |
| | GFLOPs ↓ | 716 ± 717 | N/A | **427 ± 684** | 289 (268, 310) 40% (37, 43) | N/A |

[i]Average KV Cache Length = 26     [ii]Average KV Cache Length = 1

## 6 Compute, Memory, and Storage Efficiency

Finally, we return to one of the central motivations of this paper: what kind of efficiency gains does gisting enable? To answer this question, we compare the compute requirements (CUDA wall time and FLOPs) during inference with the single-token gist models using different caching strategies:

1. **No caching**: just encoding the full prompt $t$.
2. **Instruction caching**: caching the activations of the uncompressed instruction $t$ (keys and values for all layers) into what is called the **KV cache**. This is the most common caching behavior for Transformer inference [4, 26] and is supported in libraries like Hugging Face Transformers [41]. However, it is only applicable to *decoder-only* models, since in models with bidirectional encoders like T5, the instruction representations $t$ depend on the input $x$.
3. **Gist caching**: Compressing the prompt into the gist prefix $G(t)$.

Table 3 displays the results of profiling a single forward pass through the model (i.e. one step of decoding with a single input token) with PyTorch [24] 2.0, averaged across the 252 Human instructions. Gist caching improves significantly over unoptimized models, with 40% FLOPs savings and 4-7% lower wall time for both models. Note that at these (relatively) small scales, the wall time improvements are smaller than the FLOPs reductions because much of the inference latency is caused by moving tensors from high-bandwidth memory (HBM) to the chip compute cores, i.e. what Pope et al. [26] call the "memory time". Larger sequence lengths and batch sizes will lead to additional speedups, as the overall latency becomes dominated by the actual matrix computations.

For LLaMA-7B, the picture is more nuanced when compared to caching the full instruction. Compute improvements of gist caching are smaller: a negligible decrease in FLOPs (0.11%) and a modest 1% speedup in wall time. This is because the FLOPs required for a Transformer forward pass is dominated by processing of the new input tokens, rather than self-attention with the KV cache. For example, a forward pass through LLaMA-7B with a single input token and a *2000-length* KV cache is only ~10% more expensive than the same forward pass with no KV cache—see Appendix F for more details. Nevertheless, this small decrease in FLOPs leads to a disproportionate decrease in wall time (1%), likely because the self-attention computations are slower relative to their FLOPs contribution.

At large scales and with heavily reused prompts, a 1% latency speedup can still accumulate into significant cost savings over time. More importantly, however, there are key benefits of gist caching over instruction caching besides latency: compressing 26 tokens into 1 gives more space in the input context window, which is bounded by absolute position embeddings or GPU VRAM. For example, for LLaMA-7B, each token in the KV cache requires 1.05 MB storage.[3] While the total contribution of the KV cache relative to the memory needed for LLaMA-7B inference is negligible at the prompt lengths we tested, an increasingly common scenario is developers caching many prompts across a large number of users, where storage costs quickly add up. In these scenarios, gisting allows caching of up to **26x** more prompts than full instruction caching, using the same amount of storage!

---

[3]4 (fp32 bytes) $\times$ 2 (keys+values) $\times$ 32 (num layers) $\times$ 32 (num attn heads) $\times$ 128 (head dim) = 1.05 MB.

# 7 Additional Related Work

Gisting builds upon past work in (parameter-efficient) instruction finetuning and context distillation, as discussed in Section 2. Here we will outline some additional connections to related work:

**Adapting LMs without Backprop.** As mentioned in Section 2, gisting can be viewed as a way to adapt LMs *without* gradient descent, by predicting the the prefix ($\approx$ parameters) of the adapted model. Similarly, HyperTuning [25] predicts the prefix of a model for a task using (input, output) pairs for that task. If HyperTuning is a "few-shot" adaptation method, predicting the prefix from few-shot examples, then gisting can be seen as a "zero-shot" version, predicting the prefix from the language instruction alone. Gisting also has the additional benefit over HyperTuning of being conceptually simpler: instead of training a separate LM to predict the prefix, the LM itself is used as the HyperNetwork [13], with only a tiny change to the attention masks needed for prompt compression.

**Compression and memory in transformers.** The idea of "compressing" prompts is closely related to previous attempts at storing past representations to improve memory and long-range sequence modeling in Transformers [9, 21, 27, 42, 43]. In particular, the Compressive Transformer [27] compresses transformer activations into a smaller *compressed memory* using a learned convolutional operator. Gisting can be seen as a variant of the Compressive Transformer with 3 key differences. First, the compression function is not a separately learned function, but the LM's own self-attention mechanism, controlled by an input-dependent gist token. Second, the compression function is learned jointly with instruction finetuning via the standard language modeling loss, not a specialized auxiliary reconstruction loss as in [27]. Finally, our task of interest is not long-range sequence modeling, but caching and reusing instruction following prompts for efficiency reasons.

**Sparse attention mechanisms.** By restricting attention masks, gisting draws inspiration from efficient/sparse attention methods in Transformers (see [32] for review). For example, some sliding window attention mechanisms [2, 6] may remove the need to keep the entire KV cache around during decoding, but these more general methods are not optimized for caching arbitrary parts of the input sequence of varying length, which prompt compression demands. In light of this, gisting can be viewed as an input-dependent sparse attention mechanism specifically aimed at improving efficiency of the prompting workflow now commonly used in LMs.

# 8 Discussion and Limitations

In this paper we presented gisting, a framework for prompt compression in LMs, and a simple way of implementing gist models by modifying Transformer attention masks that incurs no additional cost over standard instruction finetuning. Gisting can be seen either as a modified form of instruction finetuning or a method for (meta-)context distillation of an LM. Gist models can compress unseen OOD prompts up to 26x while maintaining output quality, resulting in up to 40% FLOPs reduction and 4.2% wall clock speedups over unoptimized models, and enabling new methods for prompt caching in encoder-decoder models. While wall-time improvements for decoder-only LMs are smaller, gisting nevertheless enables caching 1 *order of magnitude* (26x) more prompts relative to full instructions.

Gisting is a promising method for improving LM efficiency, but carries some limitations. While gisting seems to succeed in capturing the "gist" of instructions (hence the name), achieving such compression necessarily results in some loss of nuance of the original instruction; Secton 5.1 illustrates a concrete failure case we observed. Since the behavior of LMs on edge cases is already not well understood, it is *especially* important for practitioners to carefully evaluate whether the compute/accuracy tradeoff of gisting is sufficiently safe and robust for their use case, *before* deployment.

Nevertheless, we believe gisting opens several interesting directions for future work. First, the masking method presented here can be easily integrated into existing instruction finetuning workflows, but another exciting approach is to retrofit an existing, frozen LM by training a *smaller* model to compress prompts, if finetuning the larger LM is inconvenient. Second, the largest efficiency gains from gisting will result from compressing very long prompts, for example $k$-shot prompts for large $k$ that may even exceed a single context window. Finally, compression performance can likely be improved through "gist pretraining": first learning to compress arbitrary spans of natural language, before then learning prompt compression. Such objectives could be devised by inserting gist tokens into other pretraining objectives, perhaps during language modeling or T5's span corruption objective.

## Acknowledgments and Disclosure of Funding

We thank the Stanford Alpaca team, especially Xuechen Li, with assistance with Alpaca data and finetuning, and Gabriel Poesia for the codebase used to collect human evaluations. Additional thanks to Xiuyu Li for help debugging and ensuring reproducibility of the open-source codebase. JM is supported by the Open Philanthropy AI Fellowship and XLL is supported by the Stanford Graduate Fellowship.

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

# A Example PyTorch Implementation of Gist Masking

See Listing A.1 for a sample annotated implementation of gist masking. This PyTorch implementation relies on basic NumPy-style tensor operations and can thus be adapted easily to a framework like JAX.

# B Data, Training, Evaluation, and Compute Details

Code, data, and model checkpoints are available at `https://github.com/jayelm/gisting`.

**Data.** For LLaMA-7B, we used a maximum sequence length of 512 tokens during training and evaluation, except with the Human validation split, where the maximum length was increased to 768 (the Human instructions are longer). Examples longer than this length are truncated from the end. For FLAN-T5-XXL, we set a maximum input length (task $t$ + input $x$) of 128 and a maximum output length of 256, except again for the Human split, where the maximum input and output lengths were both set to 384. For both models, we set a maximum generation length of 512 tokens. These lengths were chosen such that $< 1\%$ of examples across the board were truncated during training and evaluation for both models.

**Training.** Full hyperparameters for training runs are located in Table A.1. These parameters were adapted from previous published work finetuning LLAMA/FLAN-T5. For LLaMA-7B, parameters are identical to those used in training Alpaca Taori et al. [31]. For FLAN-T5-XXL, parameters are identical to those used in training T$k$-INSTRUCT [37], except with a 5e-5 learning rate, as used in the T$k$-INSTRUCT GitHub repository,[4] rather than the 1e-5 learning rate in the paper.

LLaMA-7B was trained for 3000 steps, while FLAN-T5-XXL was trained for 16000 steps. Since there are about 130k examples in Alpaca+, given the batch sizes in Table A.1 this corresponds to about ~3 epochs and ~2 epochs of training, respsectively. These numbers, again, are identical to Taori et al. [31] and Wang et al. [36]. We note that the training time is relatively flexible; for example, we did not see substantial gains training beyond 1 epoch for FLAN-T5-XXL.

**Evaluation.** During evaluation and benchmarking, we simply greedily decoded the most likely sequence. We saw limited gains from beam search with beam size $B = 4$.

**Compute.** Experiments were run on a cluster machine with 4xA100-SXM4-80GB NVIDIA GPUs, 480GB RAM, and 16 CPUs, using PyTorch 2.0 [24], Hugging Face Transformers [41], and DeepSpeed [29]. Training runs take about ~7 hours to complete for LLaMA-7B and ~25 hours for FLAN-T5-XXL. Benchmarking results were obtained on the same machine, but using just 1 of the A100 GPUs.

Table A.1: **Hyperparameters for training runs.**

|  | LLaMA-7B | FLAN-T5-XXL |
| --- | --- | --- |
| num steps | 3000 | 16000 |
| num train epochs | $\approx 3$ | $\approx 2$ |
| batch size | 128 | 16 |
| learning rate | 2e-5 | 5e-5 |
| warmup ratio | 0.03 | 0 |
| precision | bf16 | bf16 |
| optimizer | AdamW | AdamW |
| **Deepspeed** | | |
| # GPUs (A100 80GB) | 4 | 4 |
| ZeRO stage | 3 | 3 |
| subgroup size | 1e9 | 1e9 |
| max live params | 1e9 | 1e9 |
| max reuse distance | 1e9 | 1e9 |

---

[4] `https://github.com/yizhongw/Tk-Instruct/blob/1ab6fad/scripts/train_tk_instruct.sh`

```python
import torch

def reverse_cumsum(x: torch.Tensor) -> torch.Tensor:
    """Cumulative sum from right to left.

    See https://github.com/pytorch/pytorch/issues/33520.
    """
    return x + torch.sum(x, dim=-1, keepdims=True) - torch.cumsum(x, dim=-1)

def make_mask_pre_first_gist(inputs: torch.Tensor, gist_token: int, dtype=torch.int64) -> torch.Tensor:
    """Returns a mask where all tokens prior to the first gist token are masked out.

    Args:
        inputs: a Tensor of input tokens where the last dimension is the sequence length.
        gist_token: the integer id of the gist token.
        dtype: the dtype of the mask, default int64.
    Returns:
        The requested mask.
    """
    return ((inputs == gist_token).cumsum(-1) >= 1).type(dtype)

def make_mask_post_last_gist(inputs: torch.Tensor, gist_token: int, dtype=torch.int64) -> torch.Tensor:
    """Returns a mask where all tokens after the last gist token are masked out.

    Computes the same as mask_pre_first_gist_token, but reverses the sequence before and after the cumsum.

    Args:
        inputs: a Tensor of input tokens where the last dimension is the sequence length.
        gist_token: the integer id of the gist token.
        dtype: the dtype of the mask, default int64.
    Returns:
        The requested mask.
    """
    return (reverse_cumsum(inputs == gist_token) >= 1).type(dtype)

def make_gist_mask(inputs: torch.Tensor, gist_token: int, pad_token: int, dtype=torch.int64) -> torch.Tensor:
    """Creates a gist mask from supplied inputs and gist/pad tokens.

    Tokens after the last gist cannot attend to tokens prior to the first gist. Additionally, tokens *before*
    the last gist cannot attend to tokens *after* the last gist.

    The gist mask is broadcasted to 4D (with a singleton dim 1) for compatibility with multi-headed attention
    (where dim 1 is the head dimension).

    Args:
        inputs: a Tensor of shape (batch_size, seq_len) input tokens.
        gist_token: the integer id of the gist token.
        pad_token: the integer id of the pad token. inputs == pad_token are masked out.
        dtype: the dtype of the mask, default int64.
    Returns:
        The requested mask of shape (batch_size, 1, seq_len, seq_len)
    """
    # Attention mask for tokens before the last gist token.
    pre_gist_mask = make_mask_post_last_gist(inputs, gist_token, dtype=torch.bool)[:, None, None]
    # Attention mask for tokens after the last gist token.
    post_gist_mask = make_mask_pre_first_gist(inputs, gist_token, dtype=torch.bool)[:, None, None]
    # Construct time masks by permuting to time dimension.
    pre_gist_time_mask = pre_gist_mask.permute((0, 1, 3, 2))

    mask = torch.where(pre_gist_time_mask, pre_gist_mask, post_gist_mask)
    mask = mask & (inputs != pad_token)[:, None, None]  # Mask out pad tokens.

    return mask.type(dtype)
```

Listing A.1: **Sample PyTorch implementation of gist masking.**

# C Details of ChatGPT Evaluation

```
Given a user instruction and two AI assistant responses, your job is to select the response that is
more helpful, and give reasons why. Judge responses holistically, paying special attention to whether
the response (1) correctly follows the provided instruction and (2) is factually accurate. If both
responses are equal, you may call a tie, but please use this response sparingly.

Example 1:
```
Instruction: Given the name of a city, tell me the name of the state it is in.
Input: Los Angeles
Assistant A: California
Assistant B: Wyoming
```

Your output should be:
```
{"reason": "Los Angeles is in California. Only Assistant A is correct.", "choice": "A"}
```

Example 2:
```
Instruction: Give me some examples of fun things to do with the kids on a weekend.
Assistant A: For example, what should I do with my kids on a weekend? What if I'm in a city? What if
I'm on the beach?
Assistant B: You could try going to the zoo or a museum. If you want to stay at home, you could play
board games or bake cookies.
```

Your output should be:
```
{"reason": "Assistant A doesn"t seem to be following the instruction. Assistant B provides helpful
information with examples.", "choice": "B"}
```

Example 3:
```
Instruction: Write me a python function that prints "Hello, World".
Assistant A: def hello_world(): print("Hello!")
Assistant B: "Hello, World"
```

Your output should be:
```
{"reason": "Assistant B just wrote 'Hello, World'. Assistant A actually wrote a Python function, even
if it doesn't exactly print the right thing, so overall Assistant A is better.", "choice": "A"}
```

Example 4:
```
Instruction: Translate the following sentence from English to French.
Input: I like cats.
Assistant A: Me gustan los gatos.
Assistant B: 我喜欢猫.
```

Your output should be:
```
{"reason": "Both assistants got the language wrong.", "choice": "tie"}
```

Your response should only be in the JSON format above; THERE SHOULD BE NO OTHER CONTENT INCLUDED IN
YOUR RESPONSE. Write the "reason" key before writing the "choice" key, so that you think step-by-step
before making your decision. KEEP YOUR REASONING BRIEF.
```

Listing A.2: **Full prompt given to ChatGPT for evaluation**. This prompt populates the system field in the ChatGPT API; the actual example to be evaluated is formatted like the examples in the prompt above, then given as the sole input in the user field.

We used the ChatGPT API, specifically the chatgpt-3.5-turbo engine, to run our ChatGPT evaluation experiments over a period of 2 weeks between March 27 and April 7, 2023.

The full prompt given to ChatGPT is located in Listing A.2, and contains 4 examples of desired output from ChatGPT, including preferring factually accurate responses (Example 1), preferring responses that follow the instruction, even if imperfect (Examples 2 and 3), and examples of models

Table A.2: **Pairwise Cohen's** $\kappa$ between human annotators (H1, H2, H3) and ChatGPT.

(a) LLaMA-7B

|        | H1  | H2  | H3  | Average |
|--------|-----|-----|-----|---------|
| H1     | –   | .21 | .33 | .27     |
| H2     | .21 | –   | .19 | .20     |
| H3     | .33 | .19 | –   | .26     |
| ChatGPT| .38 | .22 | .26 | .29     |

(b) FLAN-T5-XXL

|        | H1  | H2  | H3  | Average |
|--------|-----|-----|-----|---------|
| H1     | –   | .33 | .34 | .34     |
| H2     | .33 | –   | .33 | .33     |
| H3     | .34 | .33 | –   | .33     |
| ChatGPT| .35 | .18 | .34 | .29     |

being equally wrong (Examples 4). For the two models under comparison, we randomized the order of presentation of each model as either Assistant A or Assistant B, to avoid order effects.

ChatGPT was instructed to only respond in JSON format, outputting first a `reason` key followed by a `choice` key, to encourage chain-of-thought reasoning [39]. On rare occasions ($< 0.25\%$ of the time), ChatGPT would output a response that did not conform to the requested JSON format (e.g. it would just give an unstructured paragraph). In these cases we manually went through and converted these responses to JSON, without altering ChatGPT's reasoning.

In total, we collected ~22.5k judgments from ChatGPT for an estimated cost of $29.28. The full outputs for each model across the Alpaca+ validation splits, as well as ChatGPT's responses and choices, are available in the code link above.

# D   Additional Human Evaluation Details and Results

## D.1   Experimental Details

For each of the 100 examples randomly selected from the Human validation split, we recruited 3 US or UK-based, English-fluent annotators from Prolific, an online crowdsourcing platform. Experiments were IRB approved under a generic human experiments IRB given to the authors.

The annotation interface given to Prolific crowdworkers is located in Figure A.1. To verify task comprehension, participants were shown two simple examples before the main body of the task (Figure A.2), and were required to answer correctly before proceeding. We compensated participants USD $14.35/hour for an estimated cost (including Prolific fees) of USD $141.64.

## D.2   Additional Results

See Table A.2 for a breakdown of Cohen's $\kappa$ between human annotators and ChatGPT. We used a weighted version of Cohen's $\kappa$ with linear weights, since the response scale is ordinal (e.g. "tie" is a closer judgment to "pos control win" than "gist win").

# E   Exact Match Results

See Figure A.3 for a plot of exact match rates for the gist and positive control models (as measured by exact string match).

# F   Additional FLOPs details

The FLOPs required for a Transformer formward pass with varying KV cache lengths can be estimated by modifying existing equations to account for self-attention back to the KV cache. As an example, we modify the FLOPs equations used for computing FLOPs in the Chinchilla paper (Appendix F in [14]). Let **seq_len_with_past** = seq_len + kv_cache_len. Then the modified Transformer FLOPs equations are:

**Embeddings**

- $2 \times$ seq_len $\times$ vocab_size $\times$ d_model

**Judging AI Assistants**

**User Instruction:** Write a good Twitter Bio. Try to cover all the provided main points in your generated bio.

**User Input:** Name: Jane Main points: – Architect – Interested in Tech Written in first person Use casual tone

**Assistant A:** Jane is an architect who is interested in tech. She loves to solve problems and create beautiful things.

**Assistant B:** I'm Jane, an architect who loves to explore the latest tech trends. I'm always looking for new ways to make our lives easier and more convenient. Follow me for the latest tech news, tips, and inspiration!

**Which Assistant is more helpful?**

| A | B | TIE | I DON'T KNOW |

**Remember:**

When making your judgment, please consider whether the response
**1.** attempts to follow the provided instruction, and
**2.** is factually accurate.

**If you can't decide**, e.g. because the responses are the same or equally good or bad, you may call a tie, but please use this response sparingly.

**Finally**, if you do not understand the question being asked: you may still be able to make
an educated guess about which Assistant is better, for example if one assistant is clearly
not following the directions. If so, please do! Otherwise, click the
"I don't know" button. There is no need to look up additional information or
spend a huge amount of time on an example; just click "I don't know.".

Figure A.1: **Annotation interface given to Prolific crowdworkers.**

**User Instruction:** Give me some examples of fun things to do with the kids on a weekend.

**Assistant A:** For example, what should I do with my kids on a weekend? What if I'm in a city? What if I'm on the beach?

**Assistant B:** You could try going to the zoo or a museum. If you want to stay at home, you could play board games or paint photographs.

**User Instruction:** Find me a synonym of the following word.

**User Input:** speedy

**Assistant A:** quick

**Assistant B:** lazy

Figure A.2: **Example items given to humans before the start of the task.**

**Attention (Single Layer)**

- *Key, query, and value projections*: $2 \times 3 \times$ seq_len $\times$ d_model $\times$ (key_size $\times$ num_heads)
- *Key and query logits*: $2 \times$ seq_len $\times$ **seq_len_with_past** $\times$ (key_size $\times$ num_heads)
- *Softmax*: $3 \times$ num_heads $\times$ seq_len $\times$ **seq_len_with_past**
- *Softmax @ query reductions*: $2 \times$ seq_len $\times$ **seq_len_with_past** $\times$ (key_size $\times$ num_heads)
- *Final linear*: $2 \times$ seq_len $\times$ (key_size $\times$ num_heads) $\times$ d_model

**Dense Block**

- $2 \times$ seq_len $\times$ (d_model $\times$ ffw_size + d_model $\times$ ffw_size)

**Final Logits**

- $2 \times$ seq_len $\times$ d_model $\times$ vocab_size

**Total Forward Pass FLOPs**

- embeddings + num_layers $\times$ (attention_single_layer + dense_block) + final_logits

It can be seen that only 3 operations in each attention layer depend on the KV cache size, and they take up a relatively insignificant amount of FLOPs. As an illustrative example, Figure A.4 shows the relative FLOPs contributions within a single layer of attention for LLaMA-7B, assuming a *2000-length* KV cache and a single input token. Operations dependent on the KV cache constitute at most ~10% of the total attention layer FLOPs; the rest are used in KQV projections and dense layers for processing the single new input token.

Given a KV cache compression rate of 26, as observed in our Human validation split, the Chinchilla equations predict a relative improvement of Gist caching of 0.12%. This is extremely close to the 0.11% improvement actually observed in Table 3. These results show that optimizing the KV cache size does not actually lead to huge compute speedups during Transformer inference, at least for relatively small prompt lengths. Nevertheless, there are clear memory and storage benefits to be gained from prompt compression, as discussed in Section 6.

## G   Additional Examples of Model Outputs

See Listing A.3 for additional examples of instructions, inputs, model completions (for LLaMA-7B and FLAN-T5-XXL across Pos, Gist, TF-IDF, and Neg conditions), human judgments, and ChatGPT judgments (+reasons). Examples are not cherry-picked; we enumerate the first 10 examples of the Alpaca+ Human validation split.

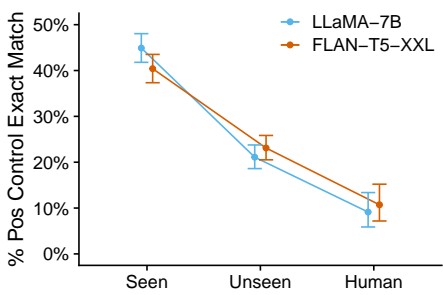

Figure A.3: **Exact match rates**. Rate at which the 1 token gist models give the same output as the positive control (exact string match). Error bars are 95% exact binomial confidence intervals.

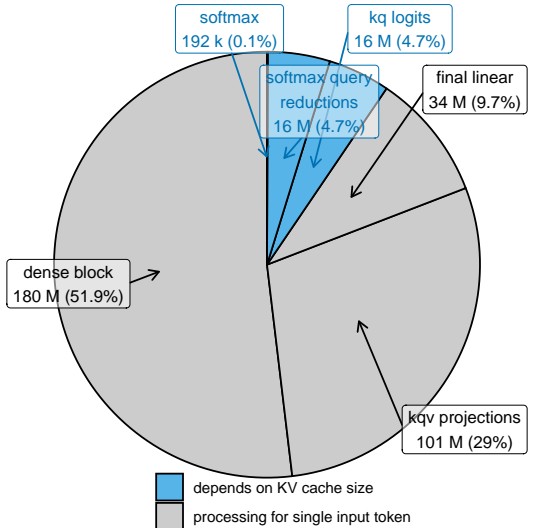

Figure A.4: **FLOPs for each operation involved in a single layer of self attention with a 2000-length KV cache**, according to the Chinchilla estimates for LLaMA-7B. At most 9.6% of FLOPs can be optimized away by reducing the size of the KV cache.

# Example 1

**Instruction:** Write a good Twitter Bio. Try to cover all the provided main points in your generated bio.  <G1>
**Input:** Name: Jane
    Main points:
    - Architect
    - Interested in Tech
    Written in first person
    Use casual tone
**Reference Output:** I like to think of myself as a kind of architect.
I'm interested in the tech side of things, and I love coming up with new ideas for how to make things better.
**TF-IDF Word:** bio

## LLaMA-7B
**Pos:** I'm Jane, an architect and tech enthusiast.
I'm interested in the latest trends and always looking to learn more.
I'm here to share my thoughts and experiences with the world.
**Gist:** I'm Jane, an architect and tech enthusiast.
I love to explore new places and learn about different cultures.
I'm always looking for fun things to do and new experiences.
I'm an avid reader and enjoy learning about the world around me.
I'm also a big fan of tech and love to stay up to date on the latest trends.
I'm looking forward to seeing what the world has to offer!
**TF-IDF:** Jane is an architect who is interested in technology.
She likes to stay up to date on the latest trends and is always looking for new ways to apply them in her work.
Jane is a creative problem solver who approaches each project with an innovative eye.
**Neg:** Jane is an architect who is interested in technology.
She likes to stay up to date on the latest trends and is always looking for new ways to incorporate them into her work.
She's an avid reader of tech news and loves to explore the latest gadgets.
Jane is also a big fan of the arts and enjoys visiting museums and galleries when she gets the chance.
**Human choice (Pos vs Gist):  Pos   Tie   Gist**
**ChatGPT choice (Pos vs Gist):  Pos**
**ChatGPT reason (Pos vs Gist):** Assistant A covers all the main points provided in the instruction, while Assistant B goes off-topic
      and doesn't mention the interest in tech as much. Assistant A also has a more professional tone, which is more appropriate
      for a Twitter bio. (Assistant A: **Pos**, Assistant B: **Gist**)

## FLAN-T5-XXL
**Pos:** I'm Jane, an architect who loves to explore the latest tech trends.
I'm always looking for new ways to make our lives easier and more convenient.
Follow me for the latest tech news, tips, and inspiration!
**Gist:** Jane is an architect who is interested in tech.
She loves to solve problems and create beautiful things.
**TF-IDF:** Architect, Tech enthusiast
**Neg:** Jane is an architect who is interested in tech.
**Human choice (Pos vs Gist):  Pos   Pos   Pos**
**ChatGPT choice (Pos vs Gist):  Pos**
**ChatGPT reason (Pos vs Gist):** Assistant A covers all the main points and uses a casual tone. Assistant B is too brief and doesn't
      provide any specific information about Jane's interests or personality. (Assistant A: **Pos**, Assistant B: **Gist**)

# Example 2

**Instruction:** Name the top cities in France that should not be missed. Include the best aspects of each place as well.  <G1>
**Reference Output:** 1- Paris, Best for: Historic sights and romantic vibes
2- Nice, Best for: Architecture and dining
3- Lyon, Best for: Dining and live music
4- Bordeaux, Best for: Wine and antiques
5- Marseille, Best for: Museums and nightlife
6- Strasbourg, Best for: Historic architecture and dining
**TF-IDF Word:** missed

## LLaMA-7B
**Pos:** The top cities in France that should not be missed include Paris, which is known for its iconic architecture and bustling
      nightlife, and Nice, which is a beautiful city on the Mediterranean known for its colorful culture and delicious cuisine.
**Gist:** The top cities in France that should not be missed are Paris, Nice, Lyon, Toulouse, Marseille, and Cannes.
The must-see attractions in each city include the Eiffel Tower in Paris, the Promenade des Anglais in Nice, the Old Town in Lyon,
      the Canal du Midi in Toulouse, the Port of Marseille in Marseille, and the Croisette in Cannes.
**TF-IDF:** I missed you.
**Neg:** The most popular social media platform in Europe is the Android-based Ovi from Nokia.
**Human choice (Pos vs Gist):  Pos   Tie   Gist**
**ChatGPT choice (Pos vs Gist):  Gist**
**ChatGPT reason (Pos vs Gist):** Assistant A provides a more comprehensive list of top cities in France and includes the best aspects
      of each place. Assistant B only mentions two cities and their general characteristics without providing specific attractions
      or details. (Assistant A: **Gist**, Assistant B: **Pos**)

## FLAN-T5-XXL
**Pos:** The top cities in France that should not be missed include Paris, Marseille, Lyon, Toulouse, Nice, and Monaco.
Paris is the capital of France and is home to the Eiffel Tower, Notre-Dame de Paris, and the Louvre Museum.
Marseille is a beautiful city with a vibrant nightlife, and Lyon is a great place to visit for its food and wine.
Toulouse is a charming city with a vibrant nightlife, and Monaco is a beautiful place to visit for its casinos and luxurious hotels.
**Gist:** Paris is one of the most beautiful cities in France.
It is a great place to visit for its architecture, culture, and history.
The city is also known for its beautiful gardens and parks, as well as its many museums and art galleries.
The Eiffel Tower is one of the most iconic landmarks in the world, and it is a must-see for anyone visiting Paris.
The Louvre is another must-see for anyone visiting Paris.
It is a stunning museum that houses some of the most iconic art pieces in the world.
The Notre-Dame de Paris is another must-see for anyone visiting Paris.
**TF-IDF:** I would be very disappointed.
**Neg:** - The Shawshank Redemption - Forrest Gump - The Godfather - Pulp Fiction - The Dark Knight
**Human choice (Pos vs Gist):  Pos   Pos   Pos**
**ChatGPT choice (Pos vs Gist):  Gist**

**ChatGPT reason (Pos vs Gist):** Assistant A only mentions Paris and provides more detailed information about the city, while Assistant B mentions several cities but doesn't provide much detail about any of them. Therefore, Assistant A is more helpful. (Assistant A: **Gist**, Assistant B: **Pos**)

## Example 3

**Instruction:** Give the news title a category. Pick a category from the list of News & Buzz, Travel, Style, Arts & Culture, Politics, Tech, and Science & Health. <G1>
**Input:** The #Banksy Exhibit in Cambridge, MA is absolutely terrific.
**Reference Output:** Arts & Culture
**TF-IDF Word:** bu

### LLaMA-7B
**Pos:** Arts & Culture
**Gist:** Arts & Culture
**TF-IDF:** No such bias detected.
**Neg:** The Banksy Exhibit in Cambridge, MA is absolutely fantastic.
**Human choice (Pos vs Gist):**   Tie    Tie    Tie
**ChatGPT choice (Pos vs Gist):**   Tie
**ChatGPT reason (Pos vs Gist):** Both assistants correctly identified the news title as related to Arts & Culture. (Assistant A: **Gist**, Assistant B: **Pos**)

### FLAN-T5-XXL
**Pos:** Arts & Culture
**Gist:** Culture
**TF-IDF:** The Banksy Exhibit in Cambridge, MA is absolutely remarkable.
**Neg:** The #Banksy Exhibit in Cambridge, MA is remarkable.
**Human choice (Pos vs Gist):**   Pos    Pos    Pos
**ChatGPT choice (Pos vs Gist):**   Pos
**ChatGPT reason (Pos vs Gist):** The news title is about an art exhibit, so Assistant B's response of 'Arts & Culture' is more accurate and specific than Assistant A's 'Culture'. (Assistant A: **Gist**, Assistant B: **Pos**)

## Example 4

**Instruction:** You will be tested on your knowledge of classic witticisms and aphorisms by completing the given aphorism. Write the original quote as the answer. <G1>
**Input:** two things are infinite
**Reference Output:** The universe and human stupidity; and I'm not sure about the universe.
(attributed to Albert Einstein)
**TF-IDF Word:** a

### LLaMA-7B
**Pos:** The number of integers and the number of witticisms.
**Gist:** "Two things are infinite, the universe and human stupidity."
- Albert Einstein
**TF-IDF:** The number of integers and the number of real numbers.
**Neg:** Two things are infinite: the number of integers and the number of primes.
**Human choice (Pos vs Gist):**   Gist    Gist    Gist
**ChatGPT choice (Pos vs Gist):**   Gist
**ChatGPT reason (Pos vs Gist):** Assistant A correctly completes the quote with the original author, while Assistant B's response is not factually accurate. (Assistant A: **Gist**, Assistant B: **Pos**)

### FLAN-T5-XXL
**Pos:** "Time and space are infinite."
**Gist:** "Actions and consequences are infinite."
**TF-IDF:** The first sentence is a statement, while the second one is an opinion.
**Neg:** The two things that are infinite are time and space.
**Human choice (Pos vs Gist):**   Pos    Pos    Pos
**ChatGPT choice (Pos vs Gist):**   Tie
**ChatGPT reason (Pos vs Gist):** Both assistants did not provide the correct completion of the given aphorism. The original quote is 'Two things are infinite: the universe and human stupidity; and I'm not sure about the universe.' Therefore, neither assistant is helpful. (Assistant A: **Gist**, Assistant B: **Pos**)

## Example 5

**Instruction:** Write the HTML syntax for the required action. <G1>
**Input:** adding an image to the page
**Reference Output:** 
**TF-IDF Word:** syntax

### LLaMA-7B
**Pos:** 
**Gist:** <nooutput>
**TF-IDF:** 
**Neg:** 
**Human choice (Pos vs Gist):**   Pos    IDK    Pos
**ChatGPT choice (Pos vs Gist):**   Pos
**ChatGPT reason (Pos vs Gist):** Assistant A provides the correct HTML syntax for adding an image to the page, while Assistant B doesn't provide any output. (Assistant A: **Pos**, Assistant B: **Gist**)

### FLAN-T5-XXL
**Pos:** 
**Gist:** 
**TF-IDF:** 
**Neg:** Adding an image to the page.
**Human choice (Pos vs Gist):**   Tie    Gist    Tie
**ChatGPT choice (Pos vs Gist):**   Tie
**ChatGPT reason (Pos vs Gist):** Both assistants provided the correct HTML syntax for adding an image to the page. (Assistant A: **Pos**, Assistant B: **Gist**)

# Example 6

**Instruction:** Write a to-do list based on the given information. `<G1>`
**Input:** Daily tasks in the pharmacy
**Reference Output:** - Police sidewalk and entrance for litter
- Assess window displays to make sure they are not outdated or faded
- Clean all doors and remove outdated and unnecessary signs
- Fill bags at each checkout area
- Clean and straighten checkout counters
- Tidy up seating/waiting area
- Face (pull product forward) in all departments
- Check one department a day for short and outdated product
- Empty all trash receptacles
- Vacuum (when the store is closed or only if customer traffic allows for it during store hours)
**TF-IDF Word:** information

## LLaMA-7B
**Pos:** - Check inventory of medications
- Reorder medications
- Prepare patient profiles
- Enter patient information into the computer system
- Verify patient information
- Monitor expiration dates of medications
- Clean and restock shelves
- Answer telephone and provide assistance as needed
- Process prescriptions
- Refill prescriptions
- Monitor inventory of supplies
- Stock shelves
- Clean and restock shelves
- Answer telephone and provide assistance as needed
**Gist:** - Check inventory and order new supplies
- Refill prescriptions
- Answer customer questions
- Process insurance paperwork
- Clean the pharmacy
- Stock shelves
- Count pills
- Take customer phone calls
- Monitor expiration dates on medications
- Update patient profiles
- Perform administrative tasks
- Review patient charts
- Contact doctors about prescriptions
- Answer questions from customers
- Process payments
- Maintain inventory
- Monitor controlled substances
- Perform quality control tests
- Review patient profiles
- Contact doctors about prescriptions
- Answer questions from customers
- Process payments
- Maintain inventory
- Monitor expiration dates on medications
- Update patient profiles
- Perform administrative tasks
- Review patient charts
- Contact doctors about prescriptions
- Answer questions from customers
- Process payments
- Stock shelves
- Count pills
- Take customer phone calls
- Monitor expiration dates on medications
- Update patient profiles
- Perform administrative tasks
- Review patient charts
- Contact doctors about prescriptions
- Answer questions from customers
- Process payments
- Maintain inventory
- Monitor expiration dates on medications
- Update patient profiles
- Perform administrative tasks
- Review patient charts
- Contact doctors about prescriptions
- Answer questions from customers
- Process payments
- Maintain inventory
- Monitor expiration dates on medications
- Update patient profiles
- Perform administrative tasks
- Review patient charts
- Contact doctors about prescriptions
- Answer questions from customers
- Process payments
- Maintain inventory
- Monitor expiration dates
**TF-IDF:** 1.

Checking expiry dates of medications
2.
Re-stocking shelves
3.
Entering prescription information into the computer system
4.
Answering customer questions
5.
Preparing and labeling prescriptions
6.
Scanning barcodes to check for potential drug interactions
7.
Processing insurance claims
8.
Maintaining inventory levels of medications
9.
Cleaning and organizing the pharmacy
10.
Updating patient profiles with new information
**Neg**: The pharmacist's daily tasks include checking and filling prescriptions, maintaining inventory, and counseling patients on the proper use of medications.
**Human choice (Pos vs Gist):** **Pos**  **Tie**  **Gist**
**ChatGPT choice (Pos vs Gist):** **Pos**
**ChatGPT reason (Pos vs Gist):** Assistant A provides a concise and comprehensive list of daily tasks in the pharmacy, while Assistant B's list is repetitive and includes irrelevant tasks like counting pills and performing quality control tests. Assistant A is more helpful. (Assistant A: **Pos**, Assistant B: **Gist**)

## FLAN-T5-XXL

**Pos**: - Check inventory - Fill prescriptions - Prepare medication - Label and package medications - Check expiration dates - Ensure all medications are in stock - Check for any errors in the inventory - Check for any problems with the medication - Check for any problems with the packaging - Check for any problems with the labeling - Check for any problems with the expiration date - Check for any problems with the packaging - Check for any problems with the labeling - Check for any problems with the packaging - Check for any problems with the packaging - Check for any problems with the
**Gist**: To-do list for pharmacy: 1.
Check inventory and stock shelves 2.
Prepare prescriptions 3.
Check inventory and stock shelves 4.
Check inventory and stock shelves 5.
Check inventory and stock shelves 6.
Check inventory and stock shelves 7.
Check inventory and stock shelves 8.
Check inventory and stock shelves 9.
Check inventory and stock shelves 10.
Check inventory and stock shelves
**TF-IDF**: 1.
Stocking shelves 2.
Filing paperwork 3.
Greeting customers 4.
Taking orders 5.
Performing administrative tasks
**Neg**: The daily tasks in the pharmacy include preparing medications, filling prescriptions, assisting customers, and maintaining inventory.
**Human choice (Pos vs Gist):** **Gist**  **Pos**  **Gist**
**ChatGPT choice (Pos vs Gist):** **Pos**
**ChatGPT reason (Pos vs Gist):** Assistant A provides a comprehensive list of daily tasks in the pharmacy, including preparing and labeling medications, checking for errors and problems, and ensuring all medications are in stock. Assistant B only lists two tasks and repeats one of them multiple times. Assistant A is more helpful. (Assistant A: **Pos**, Assistant B: **Gist**)

# Example 7

**Instruction:** You are given a paper citation, convert it to the requested citation style.  <G1>
**Input:** Chicago: Vaswani, Ashish, Shazeer, Noam, Parmar, Niki, Uszkoreit, Jakob, Jones, Llion, Gomez, Aidan N., Kaiser, Lukasz, and Illia Polosukhin. "Attention Is All You Need." arXiv, (2017). https://doi.org/10.48550/arXiv.1706.03762.
    MLA:
**Reference Output:** Vaswani, Ashish, et al.
"Attention Is All You Need."
arXiv, 2017, https://doi.org/10.48550/arXiv.1706.03762.
**TF-IDF Word:** c

## LLaMA-7B

**Pos**: Vaswani, Ashish, Shazeer, Noam, Parmar, Niki, Uszkoreit, Jakob, Jones, Llion, Gomez, Aidan N., Kaiser, Lukasz, and Illia Polosukhin.
"Attention Is All You Need."
arXiv (2017).
https://doi.org/10.48550/arXiv.1706.03762.
**Gist**: Vaswani, Ashish, Shazeer, Noam, Parmar, Niki, Uszkoreit, Jakob, Jones, Llion, Gomez, Aidan N., Kaiser, Lukasz, and Illia Polosukhin.
"Attention Is All You Need."
arXiv, (2017).
https://doi.org/10.48550/arXiv.1706.03762.
**TF-IDF**: Attention Is All You Need.
arXiv.
2017. https://doi.org/10.48550/arXiv.1706.03762.
**Neg**: Chicago: Vaswani, Ashish, Shazeer, Noam, Parmar, Niki, Uszkoreit, Jakob, Jones, Llion, Gomez, Aidan N., Kaiser, Lukasz, and Illia Polosukhin.
"Attention Is All You Need."
arXiv, (2017).
https://doi.org/10.48550/arXiv.1706.03762.
**Human choice (Pos vs Gist):** **IDK**  **Tie**  **Tie**
**ChatGPT choice (Pos vs Gist):** **Gist**

**ChatGPT reason (Pos vs Gist):** Both assistants provide correct MLA citation style, but Assistant A includes a comma after 'arXiv' which is not necessary in MLA style. (Assistant A: **Gist**, Assistant B: **Pos**)

### FLAN-T5-XXL
**Pos:** Vaswani, Ashish, Shazeer, Noam, Parmar, Niki, Uszkoreit, Jakob, Jones, Llion, Gomez, Aidan N., Kaiser, Lukasz, and Illia Polosukhin.
"Attention Is All You Need."
arXiv, (2017).
**Gist:** Chicago: Vaswani, Ashish, Shazeer, Noam, Parmar, Niki, Uszkoreit, Jakob, Jones, Llion, Gomez, Aidan N., Kaiser, Lukasz, and Illia Polosukhin.
"Attention Is All You Need."
arXiv, (2017).
https://doi.org/10.48550/arXiv.1706.03762.
**TF-IDF:** Attention is all you need.
**Neg:** Attention Is All You Need
**Human choice (Pos vs Gist):** IDK    IDK    IDK
**ChatGPT choice (Pos vs Gist):** Gist
**ChatGPT reason (Pos vs Gist):** The instruction asks for MLA citation style, but neither assistant provided the citation in MLA format. However, Assistant A provided more information such as the DOI, so Assistant A is slightly more helpful. (Assistant A: **Gist**, Assistant B: **Pos**)

## Example 8
**Instruction:** Provide a pleasant compliment based on the quality you have been given.  <G1>
**Input:** Truthfulness
**Reference Output:** Everything would be better if more people were like you.
**TF-IDF Word:** pleasant

### LLaMA-7B
**Pos:** You are remarkably truthful and honest.
**Gist:** You have a gift for being truthful.
**TF-IDF:** Truthfulness is a pleasant trait because it leads to trust, which is essential for any healthy relationship.
It also helps people build trust in their relationships with others, which can lead to better communication and understanding.
Additionally, it can help people avoid getting caught up in lies and deception, which can be damaging to relationships.
**Neg:** Truthfulness is like a lion in the forest: loud, proud, and unafraid to roar.
**Human choice (Pos vs Gist):** Pos    Pos    Pos
**ChatGPT choice (Pos vs Gist):** Pos
**ChatGPT reason (Pos vs Gist):** Both responses are good, but Assistant B's response is more detailed and specific, making it a more pleasant compliment overall. (Assistant A: **Gist**, Assistant B: **Pos**)

### FLAN-T5-XXL
**Pos:** You are incredibly honest.
**Gist:** Thank you for being so honest.
**TF-IDF:** Truthfulness is like a light that guides us through the darkness.
**Neg:** Truthfulness is like a lighthouse, providing guidance and comfort in times of trouble.
**Human choice (Pos vs Gist):** IDK    Pos    Pos
**ChatGPT choice (Pos vs Gist):** Pos
**ChatGPT reason (Pos vs Gist):** Both responses are correct and follow the instruction, but Assistant B's response is more enthusiastic and personal, making it a more pleasant compliment. (Assistant A: **Gist**, Assistant B: **Pos**)

## Example 9
**Instruction:** Make a list of adjectives that can be used to describe the given brand.  <G1>
**Input:** a creative tech startup
**Reference Output:** 1.
Innovative
2.
Creative
3.
Disruptive
4. cutting-edge
5. revolutionary
6. game-changing
7. industry-leading
8. trailblazing
9. groundbreaking
10. world-changing
**TF-IDF Word:** brand

### LLaMA-7B
**Pos:** Innovative, creative, tech-savvy, forward-thinking, dynamic, agile, fast-paced, cutting-edge, inventive, pioneering, and disruptive.
**Gist:** - Innovative
- Creative
- Ambitious
- Inventive
- Passionate
- Innovative
- Imaginative
- Visionary
- Entrepreneurial
- Optimistic
- Determined
- Persistent
- Responsible
- Thoughtful
- Respectful
- Empathetic
- Generous
- Reliable

```
- Trustworthy
- Adaptable
- Agile
- Optimistic
- Creative
- Innovative
- Imaginative
- Visionary
- Entrepreneurial
- Optimistic
- Determined
- Persistent
- Responsible
- Thoughtful
- Respectful
- Empathetic
- Generous
- Reliable
- Trustworthy
- Adaptable
- Agile
```
**TF-IDF**: TechStartup
**Neg**: A creative tech startup is like a race car: it needs a great idea, a well-tuned team and the right resources to reach the finish line.
**Human choice (Pos vs Gist):**  **Pos**   **Tie**   **Gist**
**ChatGPT choice (Pos vs Gist):**  **Pos**
**ChatGPT reason (Pos vs Gist):** Both assistants provide a similar list of adjectives, but Assistant B includes some additional adjectives that are more specific to a tech startup, such as 'tech-savvy', 'forward-thinking', 'cutting-edge', 'pioneering', and 'disruptive'. Therefore, Assistant B is more helpful. (Assistant A: **Gist**, Assistant B: **Pos**)

### FLAN-T5-XXL

**Pos**: - Innovative - Creative - Tech - Startup - Innovative - Tech - Startup - Creative - Tech - Startup - Tech - Startup - Creative - Tech - Startup - Tech - Startup - Tech - Startup - Tech - Startup - Tech - Startup - Tech - Startup - Tech - Startup - Tech - Startup - Tech - Startup - Tech - Startup - Tech - Startup - Tech - Startup - Tech
**Gist**: - Innovative - Creative - Tech - Startup - Fast - Growing
**TF-IDF**: The Creative Tech Startup.
**Neg**: "Creative Tech: Unlocking the Potential of the Modern World"
**Human choice (Pos vs Gist):**  **Pos**   **Gist**   **Gist**
**ChatGPT choice (Pos vs Gist):**  **Gist**
**ChatGPT reason (Pos vs Gist):** Assistant B's response is repetitive and not helpful. Assistant A provides a concise list of relevant adjectives. (Assistant A: **Gist**, Assistant B: **Pos**)

# Example 10

**Instruction:** Suggest a recipe for a vegan dessert.  <G1>
**Reference Output:** Honey Lime Fruit Salad
Ingredients: (for 4 servings)
1/2 lb fresh strawberry(225 g), quartered
2 kiwis, peeled and diced
2 mangoes, diced
2 bananas, sliced
1/2 lb fresh blueberry(225 g)
2 tablespoons honey
1 lime, juiced

Preparation:
Place sliced fruits in a large bowl.
In a small bowl, mix honey and lime juice.
Pour syrup over the fruit and mix.
**TF-IDF Word:** dess

### LLaMA-7B

**Pos**: Chocolate-coffee Tiramisu
**Gist**: Vegan Chocolate-Dipped Coconut Macaroons:
Ingredients:
- 1 cup all-purpose flour
- 1/2 cup coconut oil, melted
- 1/2 cup almond milk
- 1 teaspoon baking powder
- 1 teaspoon vanilla extract
- 1/2 cup coconut flakes
- 1/2 cup vegan chocolate chips

Instructions:
1.
Preheat oven to 350F.
2.
In a bowl, mix together the flour, melted coconut oil, almond milk, baking powder, and vanilla extract.
3.
Grease a baking sheet with non-stick cooking spray.
4.
In a separate bowl, mix together the coconut flakes and chocolate chips.
5.
Slowly add the dry ingredients to the wet ingredients and mix until combined.
6.
Drop spoonfuls of the batter onto the prepared baking sheet.
7.
Bake for 10 minutes, or until golden brown.
8.
Let cool before serving.
Enjoy!

**TF-IDF**: Tiramisu
**Neg**: The most popular social media platform in Europe is the Android-based Ovi from Nokia.
**Human choice (Pos vs Gist): Gist   Gist   Gist**
**ChatGPT choice (Pos vs Gist): Gist**
**ChatGPT reason (Pos vs**
     **Gist):** Assistant A provides a recipe for a vegan dessert, while Assistant B suggests a recipe that may not be vegan-friendly
     as it contains coffee and mascarpone cheese. Assistant A is more helpful. (Assistant A: **Gist**, Assistant B: **Pos**)

## FLAN-T5-XXL

**Pos**: A vegan dessert recipe is a vegan chocolate cake.
It is made with a combination of melted chocolate, coconut oil, and almond milk.
The cake is topped with a layer of chocolate chips and a layer of coconut cream.
The cake is baked in a preheated oven at 350 degrees for 30 minutes.
**Gist**: Vegan Chocolate Cake
**TF-IDF**: - Chocolate Cake - Vanilla Cake
**Neg**: - The Shawshank Redemption - Forrest Gump - The Godfather - Pulp Fiction - The Dark Knight
**Human choice (Pos vs Gist): Pos   Pos   Pos**
**ChatGPT choice (Pos vs Gist): Pos**
**ChatGPT reason (Pos vs Gist):** Assistant B provides a more detailed recipe for a vegan chocolate cake, including ingredients and
     baking instructions. (Assistant A: **Gist**, Assistant B: **Pos**)

Listing A.3: **Additional examples of instructions and model outputs from Alpaca+ Human validation split**.

