# OpenReview forum: "Learning to Compress Prompts with Gist Tokens"
_NeurIPS.cc/2023/Conference — NeurIPS 2023 poster_

### Official Review · Reviewer_nmKf · 2023-07-04

**Soundness:** 2 fair
**Presentation:** 3 good
**Contribution:** 3 good
**Rating:** 5
**Confidence:** 4

**Summary:**

Due to the computationally inefficient nature of long prompts for language models (LMs) today, this work proposes one framework to compress prompts into a smaller set of "gist" virtual tokens. Different from existing work that distills the context for a single NLP task, this work focuses on a distribution of tasks by predicting the gist tokens (via LMs learned embedding) for each task, seamlessly built upon instruction tuning. The design of their generator G would be the key to their success of the generality to gist tokens. The experiments show the effectiveness of their designs.

**Strengths:**

1. As soft prompts limit themselves in the computational efficiency, and fine-tuning is limited by the bulk nature by retraining to adapt LMs to each specific task, this work combines the strengths of these two, by learning some virtual tokens seamlessly upon the supervised instruction tuning (a common practice in current LLM deployment and serving). To me, the simple design working well for both training-time and inference-time is elegant and interesting;
2. The compression ratio looks really promising, making their technical contributions solid and interesting to some practitioners;
3. The paper is nearly very structured, and the experiments on some instruction tuning datasets look convincing.


**Weaknesses:**

I still have some confusions about its working mechanisms, regarding unseen prompts, and # of gist tokens (newly added vocabulary to the LM).

Firstly, is it learning one gist token (concatenate k of them in training) per task, and for new task, we do need to add more vocabularies? So it would be like one task one gist token?

Secondly, for unseen and seen prompts, is the training of gist tokens like "average of the context distillation loss" over all seen prompts for one particular task, and for unseen prompt, but still the same task type, you simply re-use existing gist tokens (the gist token embeddings would be updated by the cross-attention by the unseen prompts and learned embeddings in inference, though no updates)? Hope my understandings are correct.

I am happy to raise my scores if you can help me address them, and make me feel more interested.



**Questions:**

See my weaknesses part. Clarifying these in your paper clearly would at least be useful to some readers.

**Limitations:**

In addition to what they claimed in their paper,

The starting point of their proposal is significant to the community, as deploying LLMs with structured, templated prompts are becoming the common practice to a wide range of NLP, or general AI tasks, reducing the computational budgets would be beneficial to a lot of practitioners. However, their framework still require the access to the model logits in their in-context distillation loss, restricting the impacts to white-box LMs. And more precisely, how to reduce the costs for black-box LLMs is more interesting to others.

---

> ### Author Rebuttal · Authors · 2023-08-09
>
> Thank you to reviewer nmKf for their detailed and thoughtful review! We appreciate you find the paper “elegant and interesting”, with “really promising” compression ratios and “solid and interesting” technical contributions.
>
> The main concern of reviewer nmKf seems to be confusion about the gist compression mechanism, which we agree should be explained more clearly in the paper. Below we’ll answer reviewer nmKf’s questions and explain how we’ll revise the paper for clarity. Please also see our response to reviewer 3Sa1, who had some similar questions regarding the explanation of gist tokens in our paper.
>
> > Firstly, is it learning one gist token (concatenate k of them in training) per task, and for new task, we do need to add more vocabularies? So it would be like one task one gist token?
>
> **Only one additional gist token,** ***total,*** **is needed to enable gist caching, regardless of the number of tasks.** In other words, we increase the size of the LM embedding matrix by 1. **For each task, we concatenate k copies of the same gist token.** We will make this clearer by clarifying in L96 that the “k successive gist tokens in between” are **k copies of the same gist token,** and that the gist tokens are the same for each task at both train and test time.
>
> The reason why the gist token remains unchanged across tasks is because, as reviewer 3Sa1 also points out, **what changes is not the gist token or gist token embedding for each task, but rather the transformer activations computed *on top* of the gist tokens.** Providing the gist token as input to the model serves as a sort of “control signal” that encourages the model to learn to compress the prompt into the activations on top of the gist tokens. The model ideally learns to compress arbitrary prompts into the gist activations, so that it can generalize zero-shot to new prompts at test time.
>
> > Secondly, for unseen and seen prompts, is the training of gist tokens like "average of the context distillation loss" over all seen prompts for one particular task, …
>
>
> This description of the training process is mostly correct, though the loss is not “for one particular task”; the loss is the average distillation loss across all tasks, where you can think of a prompt as being equivalent to a task.
>
> > … and for unseen prompt, but still the same task type, you simply re-use existing gist tokens (the gist token embeddings would be updated by the cross-attention by the unseen prompts and learned embeddings in inference, though no updates)? Hope my understandings are correct.
>
> Here as well, we are not sure what you mean by “task type”: the unseen prompts define unseen tasks that the model needs to compress at test time.
>
> Hopefully some of the confusion here is cleared up by our answer to your first question. **At inference time, for unseen prompts, we add the exact same gist token seen during training after the new prompt.** The embedding of the gist token is the same as what was learned during training, and no updates to this embedding happen during test time. **What *does* change is the transformer activations on top of this gist embedding:** via self-attention, the Transformer computes a new unique set of activations on top of the gist tokens, that hopefully contain a compressed version of the new prompt, even though the prompt was never seen during training time.
>
> To summarize, by training to compress a wide distribution of tasks (=prompts) into the activations on top of the fixed gist tokens appended after the prompt (by averaging the context distillation loss across all training tasks), we hope that the transformer learns a compression mechanism that generalizes zero-shot to unseen tasks (=prompts) at test time.
>
> Again, please see our response to Reviewer 3Sa1 for additional ways in which we will make the description of our methods clearer in the paper, and let us know if you have further points of confusion—we are happy to continue answering any follow up questions.
>
> > However, their framework still require the access to the model logits in their in-context distillation loss, restricting the impacts to white-box LMs. And more precisely, how to reduce the costs for black-box LLMs is more interesting to others.
>
> This is a good point! To facilitate sharing and reproducibility with the scientific community, we focus our current paper on white-box, open source LMs for now, though optimizing inference costs of black-box LLM APIs is a fascinating problem.

---

> ### Author Response · Authors · 2023-08-15
> **Have we addressed your concerns?**
>
> As the discussion period is coming to an end, we would like to know if you have had the chance to read the rebuttal? It seems like the main concern of your review is regarding clarity on the methods in the paper—given that we've endeavored to explain the methods better (both in the rebuttal and in the paper), please let us know if we have addressed your concerns and if you are open to increasing your score.
>
> Thank you!

---

> > ### Comment · Reviewer_nmKf · 2023-08-16
> > **Replies from Reviewer nmKf**
> >
> > Thanks for your consistent help in clearing up my confusions. I am happy that my concerns have been addressed! I like your neat design, and I think it could be interesting to the community. So I retain my current positive ratings, and also I hope you can further improve your paper (e.g., make your technical details more clear to others) in the next round!

---

### Official Review · Reviewer_tZ74 · 2023-07-06

**Soundness:** 2 fair
**Presentation:** 3 good
**Contribution:** 2 fair
**Rating:** 5
**Confidence:** 4

**Summary:**

Prompting is the current way of using LLMs, but it occupies the context spaces. Instead of training the LLMs (e.g. fine-tuning), the paper presents a way to compress the prompt into gist tokens, which can be efficiently cached and reused. The method shows 26x compression rate, 40% FLOP reduction and wall clock time speed up.

**Strengths:**

(1) The idea is well motivated and straight-forward.
(2) The implementation through masking is well derived, simple, and well illustrated.
(3) The baselines for experiments are well proposed.
(4) The experiment results support the proposed method well (Neg < TF-IDF < Gist <Pos) and is intuitive.
(5) The failure cases are shown and well explained.

**Weaknesses:**

(1) The method achieve 4% wall-clock time reduction, which is not significant, especially when it compromises the accuracy. The claim that "longer sequence length and larger batch sizes can lead to higher speedup" (Line 267) is not shown quantatively.
(2) Compressing from 26 tokens to 1 token is not significant, compared to usually 2K context length of the model.

**Questions:**

Please address the weakness above. In particular,  when does prompt has longer sequence length? What would be the corresponding speedup?

**Limitations:**

The compression is lossy. The reviewer thinks this is significant because the benefits are not clear yet.

---

> ### Author Rebuttal · Authors · 2023-08-09
>
> Thank you to reviewer tZ74 for the detailed and thoughtful review! We appreciate that you find our method well motivated, straight-forward, and intuitive. The main concern of reviewer tZ74 is that the efficiency gains reported in the paper are “not significant”, regarding (1) the wall clock time reductions, and (2) the 26x prompt compression rate.
>
> We will respond to both claims below, but also keep in mind that **efficiency speedups are not the only contributions claimed by our paper**.
>
> ## On wall clock times
>
> Reviewer tZ74 claims that a “4% wall-clock time reduction…is not significant.” Note that Table 3 has 95% confidence intervals which show a *statistically significant* improvement in wall times, so we assume the reviewer is using the phrase “not significant” colloquially as “not important”. Respectfully, we believe that the threshold for significance is highly dependent on the use case, and that for large models, a 4% wall-clock time reduction can add up to **significant cost savings over time,** especially when LMs are deployed and used thousands of times daily.
>
> ### A highly optimized implementation of our method is out of scope for this paper
>
> For ease of experimentation and reproducibility, we implemented gisting framework in PyTorch in the popular Huggingface Transformers library (see supplement), with substantial Python logic that likely diminishes our reported wall-time improvements. **We believe an optimized implementation of gisting** would increase wall-time improvements, **but is also out of scope for a proof-of-concept paper** introducing our technique, especially given the numerous non-wall-time benefits of gisting.
>
> ## On compression ratios
>
> Reviewer tZ74 also argues that a 26x compression rate is not significant. Note there is some debate here, with reviewer nmKf describing the compression ratio as “really promising” and “interesting to some practitioners”. We have two responses:
>
> ### 1. 26x compression is an *average* compression rate and is the *maximum possible compression rate for this dataset*
>
> First, 26x is an **average compression rate** for the Alpaca+ human validation split. The prompts range from a minimum of 9 to a maximum of **117 tokens** long, i.e. **117x compression** in some cases. Although reviewer tZ74 correctly notes that these prompts tend to be shorter than an LM context window, we expect that this level of compression will allow LM users to avoid “context window exceeded” errors in many cases.
>
> Additionally, by using just a single gist token, we have achieved the **maximum possible compression rate** explorable with Alpaca+, which to our knowledge was created from the largest instruction following datasets available to us at the time of our experiments (Self-Instruct, Alpaca). Gisting may be competitive at compressing even longer prompts, but exploring this requires a larger and richer dataset that was unavailable to us at the time. We are happy to hear suggestions from reviewers for additional datasets to try.
>
> ### 2. 26x memory improvement is significant, *especially for prompt caching workflows*
>
> We'd like to highlight two clearly significant improvements of gisting for **prompt caching**, where the KV caches of common prompts are stored to speed up inference. This is one of the primary methods for speeding up LLM inference in production (see paper for citations). Within this framework, gisting offers:
>
> - **A**: **An *order of magnitude* decrease in storage and memory costs.** As stated in L284, gisting allows developers to cache up to **26x more prompts from users** than full instruction caching, using the same amount of storage. Even though the relative memory requirement of a *single prompt* is insignificant compared to the memory required for an LLM, if a developer wishes to have hundreds or thousands of prompts cached in GPU VRAM for fast decoding, the prompt storage requirements quickly dominate. For example, the average memory required to store the KV cache of a prompt in the human validation split for LLaMA-7B is 27.3 MB (see L281 in the paper), so caching 100 or 1000 prompts requires **2.73 or 27.3 GB VRAM**, respectively. In such cases, gisting *greatly* increases the number of possible prompts that can be simultaneously cached. We believe this is an important contribution, even if one believes the 26x reduction in prompt length is insignificant for a single prompt.
> - **B**: **New options for prompt caching in encoder-decoder models.** As stated in L258, gisting enables a form of **prompt caching** that is not possible in ordinary encoder-decoder models, since the encoder normally expects to perform bidirectional attention between the full instruction and the input. We believe this is a valuable contribution which opens up new workflows for encoder-decoder LMs.
>
> **In fact, we believe A and B above stand by themselves as valuable contributions to the community, even if there were no other reported efficiency benefits (e.g. wall-time).** We will make it clearer in the paper that these memory improvements are just as important as wall-time, expanding on L284-285.
>
> **Overall,** while it is accurate to describe gisting as lossy, our empirical results show an often negligible impact on downstream task performance. Given the numerous improvements discussed above, we believe gisting will be a useful option to developers as they consider various tradeoffs between accuracy and efficiency in LM inference.
>
> ## Other paper contributions besides efficiency speedups
>
> Finally, please keep in mind that efficiency reductions are not the only reported contributions of our paper. We also provide:
>
> - A mathematical framework for “meta-context distillation” of a language model (Section 2.1);
> - A novel way to learn compression in transformers via token-dependent attention masking (Section 3).
>
> We believe that these aspects of the paper are also stand-alone, useful contributions for those interested in memory, compression, and efficiency in transformers.

---

> ### Author Response · Authors · 2023-08-15
> **Have we addressed your concerns?**
>
> As the discussion period is coming to an end, we would like to know if you have had the chance to read the rebuttal? Please let us know whether we've addressed your concerns—we are happy to discuss more. We hope we've highlighted a variety of benefits of our method that are not explicitly concerned with wall time or the compression of a single prompt.
>
> Thank you!

---

> > ### Comment · Reviewer_tZ74 · 2023-08-16
> >
> > Thanks for the great rebuttal! I will raise my score to positive ones. Please considering accepting this paper for all the strengths illustrated.
> >
> > However, I am not convinced 4% speedup is significant. Usually, a good system paper will claim at least 15% speedup. Please consider further optimizing this.

---

### Official Review · Reviewer_W7yi · 2023-07-06

**Soundness:** 2 fair
**Presentation:** 4 excellent
**Contribution:** 2 fair
**Rating:** 6
**Confidence:** 3

**Summary:**

The authors tackle the problem of wasted compute and waste context window space from repeatedly encoding a prompt.

The authors propose gisting, in which gist tokens are inserted after the prompt, and the attention mask is modified such that tokens after the gist tokens cannot attend to tokens before the gist tokens. This forces the gist tokens to encode and compress the prompt.

Experiments are conducted with two models, LLaMA 7B and FLAN-T5-XXL 11B using a dataset the authors create from Self-Instruct and Stanford Alpaca. A hold-out set of 1000 seen, 1000 unseen, and 252 human prompts are used for evaluation.

The evaluation is conducted using a ROGUE-L score, ChatGPT3.5 model evaluation, and human evaluation score conducted on a subset of the human prompts.

Compared to an “upper-bound” of training a model with a single gist token, but no modifications to the attention mask, gisting scores near this “upper-bound” using both ROGUE-L and ChatGPT3.5.

Human evaluation results show gisting winning 52.3% and 40.6% of the time over the “upper-bound” baseline for LLaMA and FLAN-T5-XXL respectively.

Compression rates from gisting are material compared with no caching strategy, but more modest compared with caching the full instruction.


**Strengths:**

Gisting is simple to implement, requiring only adding one or more gist tokens, and a few modifications in attention masks

Gisting learns prompt compression and instruction following at the same time, incurring no additional training

Gisting can be used even with unseen prompts and has some generalization capability

Experiments use relatively large models and a relatively large dataset

Paper is clearly written

**Weaknesses:**

The evaluation scheme is the primary weakness of the paper. Human evaluators are critical given the nature of the tasks in the dataset (which consist of many open-ended generation tasks) to check how reliable ChatGPT (along with the prompt the authors present in the Appendix) is as an evaluator. We see that human evaluators have low agreement (kappa of 0.24 and 0.33, Table 2). The overall experimental results would be more convincing if human agreement was high. For example, it would be interesting to see results on a high human agreement subset (say tasks with clear objective answers) to see if Gist’s Win % over Pos was still near 50%. Furthermore, Negative Control scores ~25-30% using ChatGPT on various validation splits, which is high considering that ~59% of the tasks in Alpaca+ have no input, and the task is the only information used to generate the output. It is plausible that human evaluators have very high agreement on those outputs generated with Negative Control, but with much lower scores than ChatGPT scores, which would indicate ChatGPT + author’s prompt as an unreliable evaluator. Presented as is, it is hard to tell whether 1) the gisting process is doing well at compressing the task t, resulting in a similar performance as using the original prompt or 2) there is a large amount of randomness inherent in the evaluation process (from factors such as differing human preferences), resulting in a near 50% score (as measured by ChatGPT). These observations, along with the odd fact that reducing the compression rate does not materially improve the scores as measured by ChatGPT casts into doubt the experimental results.

As Section 5.1 shows, for some tasks, specific details need to be preserved to successfully accomplish the task. However, the authors claim that increasing the number of gisting tokens k does not help with performance. Furthermore, the choice of the number of gisting tokens is fixed for all tasks prior to training. Therefore, this method does not allow for dynamically trading off compression levels and model output quality.  It would be interesting to see evaluation results on tasks requiring differing levels of specificity.

The efficiency gains of Gisting is modest compared to instruction caching


**Questions:**

Could you please clarify “having too many gist tokens hurts performance in some cases … the increased capacity enables overfitting to the training distribution”. In the extreme case, having enough gist tokens to copy the longest prompt t, we would expect equal performance between gisting and using original task instructions t.

Could you please clarify the tie-breaking process referenced in this sentence: “Due to the subjectivity of breaking ties in a forced-ranking task…”. This is fairly important given that this is a proposed explanation for the low agreement numbers. Based on the prompt given to ChatGPT, instructions given to human evaluators, and evaluation examples found at the end of the Appendix, it appears no tie-breaking has occurred.

**Limitations:**

Gisting loses some nuance in the original instructions.

The compression level is not dynamic relative to the task.

---

> ### Author Rebuttal · Authors · 2023-08-09
>
> Thank you to reviewer W7yi for the detailed and careful review!
>
> ## On low inter-annotator agreement
>
> > The overall experimental results would be more convincing if human agreement was high.
>
> **On the contrary, when comparing two models of equal quality, we do not expect high inter-annotator agreement!** We will explain by answering this question:
>
> > Could you please clarify the tie-breaking process: “Due to the subjectivity of breaking ties in a forced-ranking task…”.
>
> Good question; we will revise this line for clarity. What we mean is that due to the subjectivity of open-ended generation tasks, there will be noisy reasons why an annotator prefers one response to another, **even if the responses come from models that are equally good.** We expect this to be true even for tasks with objective answers, since in open-ended settings, an answer can be phrased in several ways and annotators might prefer one wording over another.
>
> In our experiments, annotators have the option to call a tie, and our agreement metric takes ties into account, but neither humans nor ChatGPT call ties often (27% for humans and 13% for ChatGPT). Thus our evaluation is basically a forced choice task, where even if responses are equally good, annotators will often arbitrarily choose one response over another.
>
> **We ran a follow up experiment to verify this by asking 3 additional human annotators to rate *two different sets of samples from the LLaMA positive control***, using the same setup in the paper. As expected, the average win rate of one set of samples over the other is 50% according to humans (52%, 48%, 50%) and ChatGPT (54%). **Importantly, the average pairwise agreement of the humans in this experiment is low, and similar to the paper: a kappa of 0.30.** Meanwhile the agreement of ChatGPT with humans is 0.37. Please see the full Table in the general rebuttal PDF, and recall Table A.2 (a) in the paper shows an average agreement of **.24** for humans and **.29** for ChatGPT for LLaMA.
>
> This means **we expect low inter-annotator agreement when models are hard to differentiate.** This supports the idea that ChatGPT is similar to human annotation, and that despite low annotator agreement, the win rates across human(s) and ChatGPT are similar, and converge on gisting being similar to the positive control in many cases.
>
> As a final point, we stress that open-ended LM evaluation is challenging, and the field lacks common evaluation standards. Our forced-choice eval is similar to those done by the LMSys Chatbot Arena and recent LM projects such as Alpaca, Koala, and Vicuna. To mitigate these challenges, we used a wide spectrum of evaluations: first, lexical overlap compared to a gold response (ROUGE-L); second, AI annotation (via ChatGPT) across 1000s of outputs; third, human annotation. Together, we believe these signals provide converging evidence that gisting is competitive with positive controls, even if the inter-annotator agreement is low by design.
>
> ## Should more gist tokens improve performance?
>
> > In the extreme case, having enough gist tokens to copy the longest prompt t, we would expect equal performance between gisting and using original task instructions t.
>
> **It is not the case that you would expect equal performance to the positive control when there are many gist tokens.** This is because gist compression is an ***entirely new model capability*** that the LM has to learn **almost completely from scratch**, since the new gist embedding is randomly initialized and the LM has never seen gist masking before.
>
> The LM needs to learn a new, internal “model” $G(t)$ to compress prompts, and the number of gist tokens determines the “capacity” of this new internal model (more gist tokens = more gist activation parameters). **This means that the generalization performance of the compressor $G$ is subject to the same generalization tradeoffs we consider in standard machine learning:** more gist tokens (“a bigger model”) may result in lower training error (better compression of seen prompts) but worse test error (poorer OOD generalization).
>
> Given a finite training set, in the extreme case, a model with enough gist tokens could learn to **memorize** each training prompt with a completely unique prefix, matching the positive control during training but failing to generalize.
>
> Reviewer W7yi is correct that if the model learned to copy the prompt activations into the gist activations, and attend to the gist activations as if they were the original prompt activations, then this would result in equal performance as the positive control. **However, our results show that models do not learn this mechanism through optimization pressure alone.**
>
> ## Other points
>
> > Negative Control scores ~25-30%...which is high considering that ~59% of the tasks in Alpaca+ have no input
>
> We will clarify in the paper that the 59% figure refers to the **overall distribution** of Alpaca+. For evaluation, **we hold out prompts with non-empty inputs** (except for the Human split). **All prompts in the seen and unseen splits have inputs;** 83% of the prompts in the human split have inputs. See L126-128, though we will further clarify the unseen prompts also have inputs. This means that the negative control can usually make an educated guess about the task given the input. We believe that the fact the negative control is preferred sometimes demonstrates learnable biases in the distribution of Alpaca+, rather than issues with our evaluation.
>
> > does not allow for dynamically trading off compression levels and model output quality.
>
> We believe the problem of dynamically estimating how much compression is possible for a prompt is a fascinating one, but we leave this for future work.
>
> > efficiency gains of Gisting is modest compared to instruction caching.
>
> Please see our response to reviewer tZ74. Briefly:
>
> - Gist caching is **1 OoM more memory efficient** than instruction caching.
> - Instruction caching is not possible for encoder-decoder models.

---

> > ### Comment · Reviewer_W7yi · 2023-08-15
> >
> > Thank you for the response. My concerns related to evaluation are mostly addressed. I still think it is far more convincing to add as a supplement experiments that use a dataset with objective answers (e.g. a multiple choice dataset) and compare raw scores for gisting vs baselines without using model evaluation.
> >
> > However, after considering the authors' response, I believe the evidence weighs in favour of the author's claim that "gisting being similar to the positive control in many cases". I think the paper should be accpeted and increase my score to 6.

---

> > > ### Author Response · Authors · 2023-08-15
> > > **Thanks**
> > >
> > > Thanks to the reviewer for updating their score and for the additional feedback!
> > >
> > > We agree that an MC choice evaluation would be interesting—one of the reasons we did not explore such an eval is because it is likely OOD for the models trained in our paper, since MC often requires remembering verbatim details in the input (e.g. the choices), which the gist models are not optimized to do (as we show in our limitations). It would be interesting to explicitly explore training the on existing MC datasets (so that gist models are encouraged to remember relevant details), and perhaps this might even improve overall performance on open-ended generation tasks, but we have not yet explored this avenue due to compute constraints.

---

### Official Review · Reviewer_3Sa1 · 2023-07-07

**Soundness:** 4 excellent
**Presentation:** 3 good
**Contribution:** 3 good
**Rating:** 7
**Confidence:** 4

**Summary:**

This paper presents "gisting", which learns language models to compress instruction prompts into smaller sets of compressed context -- that includes special '<GIST>' tokens and the activation stacks above these tokens. Compressing instruction prompts allows saving context windows and saving compute for encoding instruction prompts. The task settings that this paper considers include:
* [instruction prompt][input][output] --> [compressed instruction prompt][input][output]
* [instruction prompt][output] --> [compressed instruction prompt][output]
The experimental results show that it is indeed possible to represent instruction prompts with much shorter compressed context while having only marginal performance loss. The authors evaluated the gisting model against uncompressed model on instruction following tasks comprehensively with ROUGE-L, preference of ChatGPT, and human evaluation.

**Strengths:**

* Novelty: compressing instruction prompts to save context windows and save compute for encoding instruction prompts is a novel and useful idea that can have good impact on real-world applications.
* Very solid and well-designed experiments and evaluations: the authors present comprehensive evaluations by ROUGE-L (which is lexical), ChatGPT (which can be considered an automatic semantic evaluation), and human. Furthermore, when showing the human evaluation (Table 2), which is based on a subset of evaluation set, it is compared with ChatGPT results side-by-side and shows that outcomes are consistent and inter-annotator agreement level is similar.
* Writing clarity: the paper is very easy to follow in most of the part. Figures, plots, and tables are well-polished.

**Weaknesses:**

There are some descriptions in the paper such as **G(t) will be a set of soft gist tokens smaller than the number of tokens in t ...** (line 66-68), **Gisting compresses prompts into “gist tokens”** (Figure 1 caption), etc, suggesting that the gist **tokens** contains compressed information. However, if I understand the paper and code correctly, a gist token is really just a special additional token <GIST> in vocabulary and the gist token embedding does not change according to task t. What really contains the compressed information is those activations above the gist tokens.

In other words, gisting compresses prompts into gist "activations" instead of gist "tokens" (are inputs to the model). G(t) are activations, not "soft gist tokens". When I think of soft tokens, I would think of the type of soft tokens as in prompt tuning, and this seems clearly different. Please correct me if I'm wrong but I think these descriptions are incorrect and misleading and need to be fixed before the paper can be published.

**Questions:**

How is compression factor (in figure 3) estimated? It doesn't seem to be explained in the paper.

**Limitations:**

Limitations are properly discussed in the conclusion section.

---

> ### Author Rebuttal · Authors · 2023-08-09
>
> Thank you to reviewer 3Sa1 for their detailed and helpful review! We are glad you find the paper “novel and useful”, with “very solid and well-designed experiments and evaluations”, as well as “very easy to follow”. Here are some responses to your comments and questions:
>
> > However, if I understand the paper and code correctly, a gist token is really just a special additional token <GIST> in vocabulary and the gist token embedding does not change according to task t. What really contains the compressed information is those activations above the gist tokens.
>
> Thank you for pointing this out! Your understanding is correct. We agree that we should clarify in the paper that the goal is to compress prompts into gist activations, rather than gist tokens, of which there is a single gist token embedding that remains unchanged for each task. An analogous way to think about gisting is that it compresses prompts into a soft KV prefix (a la prefix-tuning) rather than soft tokens (a la prompt tuning). We will make the following changes in the camera-ready version of the paper, if accepted:
>
> - L6-7 (abstract): change “gist tokens” to “transformer activations on top of ‘gist’ tokens”
> - L31: Change “gist tokens” to “transformer activations on top of ‘gist’ tokens”
> - Figure 1 caption: change “compresses prompts into ‘gist tokens’” to “compresses prompts into transformer activations on top of ‘gist’ tokens”
> - L67: Change “will be a set of soft gist tokens” to “will the transformer activations on top of a set of gist tokens”
> - L100: Replace “forces the model to compress the information in the prompt into the gist tokens” to “forces the model to compress the information in the prompt into the gist prefix”
> - L142: change “failed to compress any information into the gist tokens” to “failed to compress any information into the gist prefix”
> - L185: change “compressing prompts into a single token” to “compressing prompts into a single token’s worth of activations”
> - L260: Replace with “Caching the compressed activations G(t) on top of the gist tokens”
>
> Also see our response to reviewer nmKf, who has similar questions about the paper.
>
> > How is compression factor (in figure 3) estimated? It doesn't seem to be explained in the paper.
>
> Thank you for pointing this out. The compression factor is calculated by computing the average length (in tokens) of the instruction in each validation split and dividing by the number of gist tokens in the model (since we are replacing the number of tokens in the prompt with N gist tokens). For a single gist token model, a 26x compression factor implies that the average length of the instructions in the human evaluation set is 26 tokens long, as stated in L130. We will make this clearer by adding a sentence after L189 in the paper.

---

> > ### Author Response · Authors · 2023-08-15
> > **Have we addressed your concerns?**
> >
> > As the discussion period is coming to an end, we would like to know if you have had the chance to read our rebuttal? Please let us know if we have addressed all of your concerns in the initial review, and if you have any additional follow-up questions!
> >
> > Thank you!

---

> > > ### Comment · Reviewer_3Sa1 · 2023-08-15
> > >
> > > The rebuttal responses to all the reviewers look good to me. I don't have further comments at this time and still recommend accepting this work (changing score from 6 to 7).

---

### Author Rebuttal · Authors · 2023-08-09

Thank you to reviewers 3Sa1, W7yi, tZ74, and nmKf for their uniformly detailed and constructive reviews, and to the area chair for overseeing this process!

We are glad that a majority of the reviewers are currently positive on the paper, and that reviewers found our ideas “novel and useful” (3Sa1), “well motivated and straight-forward” (tZ74), “simple to implement” (W7yi), and “elegant and interesting” (nmKf), and our experiments “very solid and well-designed” (3Sa1), “convincing” (nmKf) and “support[ing] the proposed method well” (tZ74). We are especially appreciative that reviewers find the paper “easy to follow” (3Sa1), clearly written (W7yi), and “very structured” (nmKf).

We have responded individually to each reviewer’s concerns in direct replies.

**Our attached rebuttal PDF contains a table of inter-annotator agreement statistics we observed from an experiment run in response to reviewer W7yi;** please see that review for additional context. For convenience, we have also reprinted the inter-annotator agreement statistics reported in the original paper as well.

---

### Decision · Program_Chairs · 2023-09-21

**Decision:**

Accept (poster)

**Comment:**

This paper introduces a novel concept of compressing instruction prompts to improve real-world applications. The experiments and evaluations are well-designed, including lexical, automatic semantic, and human evaluations. The writing is clear and easy to follow. However, one reviewer pointed out that the claimed 4% time reduction may not justify the compromise in accuracy (+ instruction caching seems to be a strong baseline already). Also, low human evaluator agreement and uncertainties about ChatGPT's reliability.

Several points need to be addressed before publication:
- The paper contains misleading descriptions regarding "gist tokens" and their role in compressing prompts.
- Better to show generalize ability of prompt compression in a longer prompt context, for example 2k content length.
- Make the technical details more clear, especially about the unseen prompts.